# Oncolytic virotherapy induced CSDE1 neo-antigenesis restricts VSV replication but can be targeted by immunotherapy

Timothy Kottke[1], Jason Tonne[1], Laura Evgin[1], Christopher B. Driscoll [1], Jacob van Vloten [1], Victoria A. Jennings[2,3], Amanda L. Huff[1], Brady Zell[1], Jill M. Thompson[1], Phonphimon Wongthida[1], Jose Pulido[1], Matthew R. Schuelke [1], Adel Samson[3], Peter Selby[3], Elizabeth Ilett [3], Mark McNiven[4], Lewis R. Roberts [4], Mitesh J. Borad[5], Hardev Pandha[6], Kevin Harrington [2], Alan Melcher[2] & Richard G. Vile [1,3,7✉]

In our clinical trials of oncolytic vesicular stomatitis virus expressing interferon beta (VSV-IFNβ), several patients achieved initial responses followed by aggressive relapse. We show here that VSV-IFNβ-escape tumors predictably express a point-mutated CSDE1$^{P5S}$ form of the RNA-binding Cold Shock Domain-containing E1 protein, which promotes escape as an inhibitor of VSV replication by disrupting viral transcription. Given time, VSV-IFNβ evolves a compensatory mutation in the *P/M* Inter-Genic Region which rescues replication in CSDE1$^{P5S}$ cells. These data show that CSDE1 is a major cellular co-factor for VSV replication. However, CSDE1$^{P5S}$ also generates a neo-epitope recognized by non-tolerized T cells. We exploit this predictable neo-antigenesis to drive, and trap, tumors into an escape phenotype, which can be ambushed by vaccination against CSDE1$^{P5S}$, preventing tumor escape. Combining frontline therapy with escape-targeting immunotherapy will be applicable across multiple therapies which drive tumor mutation/evolution and simultaneously generate novel, targetable immunopeptidomes associated with acquired treatment resistance.

[1] Department of Molecular Medicine, Mayo Clinic, Rochester, MN, USA. [2] Chester Beatty Laboratories, Division of Radiotherapy and Imaging, The Institute of Cancer Research, London, UK. [3] Leeds Institute of Medical Research, University of Leeds, Leeds, UK. [4] Division of Gastroenterology and Hepatology, Mayo Clinic, Rochester, MN, USA. [5] Division of Hematology/Oncology, Mayo Clinic, Scottsdale, AZ, USA. [6] Faculty of Health and Medical Sciences, University of Surrey, Guildford, UK. [7] Department of Immunology, Mayo Clinic, Rochester, MN, USA. ✉email: vile.richard@mayo.edu

Escape from frontline therapy is a major cause of treatment failure in cancer patients[1–4], wherein a subset of patients initially develop promising clinical responses, followed by aggressive, lethal, tumor growth. Hence, strategies that reduce treatment failure through tumor escape would be highly significant.

We have shown that treatment-escaped tumors differ significantly from primary tumors immunogenically, genetically, and phenotypically[1–12], due to mutational plasticity and selection of treatment-resistant clones[1–4,9–11]. These treatment-escape phenotypes arise, at least in part, through the evolution of a pool of mutated tumor cells from which highly aggressive, treatment-resistant clones are rapidly selected[1–4,9–11]. In this respect, APOBEC (apolipoprotein B mRNA editing enzyme, catalytic polypeptide-like) cytosine deaminases provide an endogenous source of DNA mutation-driving cancer evolution in response to a variety of different frontline treatments[1,9–11,13–20]. APOBEC3 cytosine deaminases act as innate antiviral restriction factors and catalyze cytosine to uracil deamination of ssDNA (C–T transitions and C–G transversions)[1,16,17]. Although the human genome encodes seven APOBEC3 enzymes (3A–H), the mouse encodes a single gene, mAPOBEC3[1,17,21], which has similar activities to those of human APOBEC3B (hAPOBEC3B)[10]. Consistent with APOBECs as drivers of tumor escape, the APOBEC3B signature is associated with therapeutic resistance in multiple cancers[1,14,15,18–20].

We have shown that APOBEC3 induction following frontline treatment with adoptive T-cell therapy, chemotherapy, or oncolytic virotherapy has profound consequences for the generation of escape variants from all three types of therapy[9–11]. With respect to oncolytic virotherapy, we, and others, have developed vesicular stomatitis virus (VSV), a single-strand negative sense RNA virus (Rhabdovirus, Indiana serotype), as an oncolytic platform for clinical testing[22–31]. VSV, which is highly sensitive to inhibition by interferon (IFN), shows selective replication in Type I IFN-defective tumor cells, while being rapidly shut down in (IFN-responsive) normal cells[22–24]. For the clinical development of VSV[32], we overexpressed the IFNβ gene in the virus[25] to enhance safety (increased antiviral IFN expressed in normal cells) and to increase the immunogenicity of infected/dying tumor cells, as IFNβ acts as a key signal 3 cytokine to facilitate priming of tumor-reactive T cells[25,32,33].

In our clinical trials of VSV-IFNβ as an oncolytic[22–31], several patients achieved initial responses followed by aggressive escape. To understand the mechanistic basis of these effects, we established in vitro and in vivo models in which suboptimal levels of VSV infection leads to the escape of virus-resistant cells[9,10]. We showed that suboptimal infection with VSV induced Type I IFN-dependent hAPOBEC3B- or mAPOBEC3-induced mutation of the cell genome, degradation of the viral genome, and escape of virus/oncolysis-resistant (VSV-ESC) cells[9,10]. VSV-ESC cells[10] carried stable APOBEC3B mutational signatures in multiple genes[11], some of which might be critical for escape. Simultaneously, we reasoned that some of these mutations may also induce neo-antigenesis—the generation of neo-epitopes with increased major histocompatibility complex (MHC) binding and immunogenicity[34–36], rendering VSV-ESC cells susceptible to T-cell attack. Consistent with this hypothesis, we identified a C–T mutation in the CSDE1 gene (CSDE1^C-T) (proline to serine at αα5, CSDE1^P5S)[11] in VSV-ESC cells[10,11], which generated a heteroclitic[37–39] neo-epitope, which primed T-cell responses against both itself (CSDE1^P5S) and, to a lesser extent, wild-type CSDE1^[WT11]. CSDE1, is multi-functional RNA-binding protein that regulates RNA translation[40–47].

We show here that CSDE1 is a critical mediator of VSV replication, that the CSDE1^P5S mutation facilitates escape by inhibiting the oncolytic activity of the virus, and that neo-antigenesis of CSDE1^WT to CSDE1^P5S generates an Escape-Associated Tumor Antigen (EATA), which can be ambushed by vaccination. Therefore, it is possible to combine frontline virotherapy with escape-targeting immunotherapy, to target the evolving immunopeptidome of treatment-resistant tumor cells.

## Results

**Escape from VSV-IFNβ oncolysis is associated with high-frequency mutation in CSDE1.** B16 populations, which we had previously investigated as targets for virus-mediated treatment escape (ESC) through APOBEC3 mutagenesis, selected for escape from VSV-GFP (B16-VSV-GFP-ESC) were heterogeneous for both CSDE1^WT and CSDE1^C-T (Supplementary Fig. 1A, B). B16-HSVtk cells, which escaped ganciclovir chemotherapy[9], had no mutation in CSDE1 (Supplementary Fig. 1C). We had shown that expression of IFNβ from the virus increased IFN, APOBEC3, and the number of virus-resistant cells[10]. Whereas CSDE1^C-T was present at ~50% in B16-VSV-GFP-ESC cells, over 90% of CSDE1 sequence in B16-VSV-IFNβ-ESC cells was CSDE1^C-T, suggesting mutation at most of the alleles in ESC cells (Supplementary Fig. 1D). However, when both B16-VSV-GFP-ESC and B16-VSV-IFNβ-ESC cells were selected from B16 cells expressing short hairpin RNA (shRNA) against mAPOBEC3[10], only CSDE1^WT was present (Supplementary Fig. 1E, F). CSDE1^C-T was present at >90% in murine and human VSV-IFNβ-ESC cells, and always at a higher clonality than in VSV-GFP-ESC cells (Supplementary Fig. 1G–I). CSDE1^C-T was present in Mel888 tumors, which escaped VSV-hIFNβ in vivo[10], but only in ~30–50% of the cells (Supplementary Fig. 1J), probably reflecting less efficient in vivo infection.

Taken together, these data are consistent with the CSDE1^C-T mutation, which has a typical mAPOBEC3/APOBEC3B signature (TTCA-TCCA)[11,14,48], being induced through Type I IFN induction of mAPOBEC3/hAPOBEC3B activity[10,11] at a high clonality in VSV-IFNβ ESC cells across species and tumor types (Supplementary Fig. 1).

**CSDE1 is a positive mediator of VSV replication and oncolysis.** These data suggested that CSDE1 may be critical for the replication/oncolytic activity of VSV, that CSDE1^C-T mutation drives escape, and that co-expression of IFNβ enhances mutation of this escape-promoting gene. Consistent with this, replication of (Fig. 1A–C), and oncolysis by (Fig. 1D), VSV-GFP was reduced by >2 orders of magnitude by CSDE1 knockdown in human cells[46].

**CSDE1^C-T inhibits VSV replication.** Similarly, VSV-IFNβ replicated to significantly higher titers in B16 cells over-expressing CSDE1^WT (p < 0.0001 at 72 h) (Fig. 1E), but significantly worse in B16 cells overexpressing CSDE1^C-T (p < 0.0001 at 72 h), compared to parental B16 (Fig. 1E). B16-CSDE1^C-T cells still have both normal alleles of CSDE1 in situ and express endogenous CSDE1^WT, showing that CSDE1^P5S could still exert a strong selective pressure on the viral genome, across species and histological types. Multiple passage of VSV-IFNβ through human Hep3B-CSDE1^WT (as a model of human hepatocellular cancer cells against which we are testing VSV-IFNβ in clinical trials) increased replication compared to passage through Hep3BP parental cells (Fig. 1F). In contrast, after just a single passage through Hep3B-CSDE1^C-T cells, titers were significantly lower than with passage through Hep3BP (p < 0.0001) (Fig. 1F). By passage 3 through Hep3B-CSDE1^C-T, titers began to recover and reached almost Hep3BP levels by P5 (Fig. 1F). Virus recovered from five passages through Hep3BP (from Fig. 1F), replicated well on Hep3BP cells (Fig. 1G), but had orders-of-magnitude lower

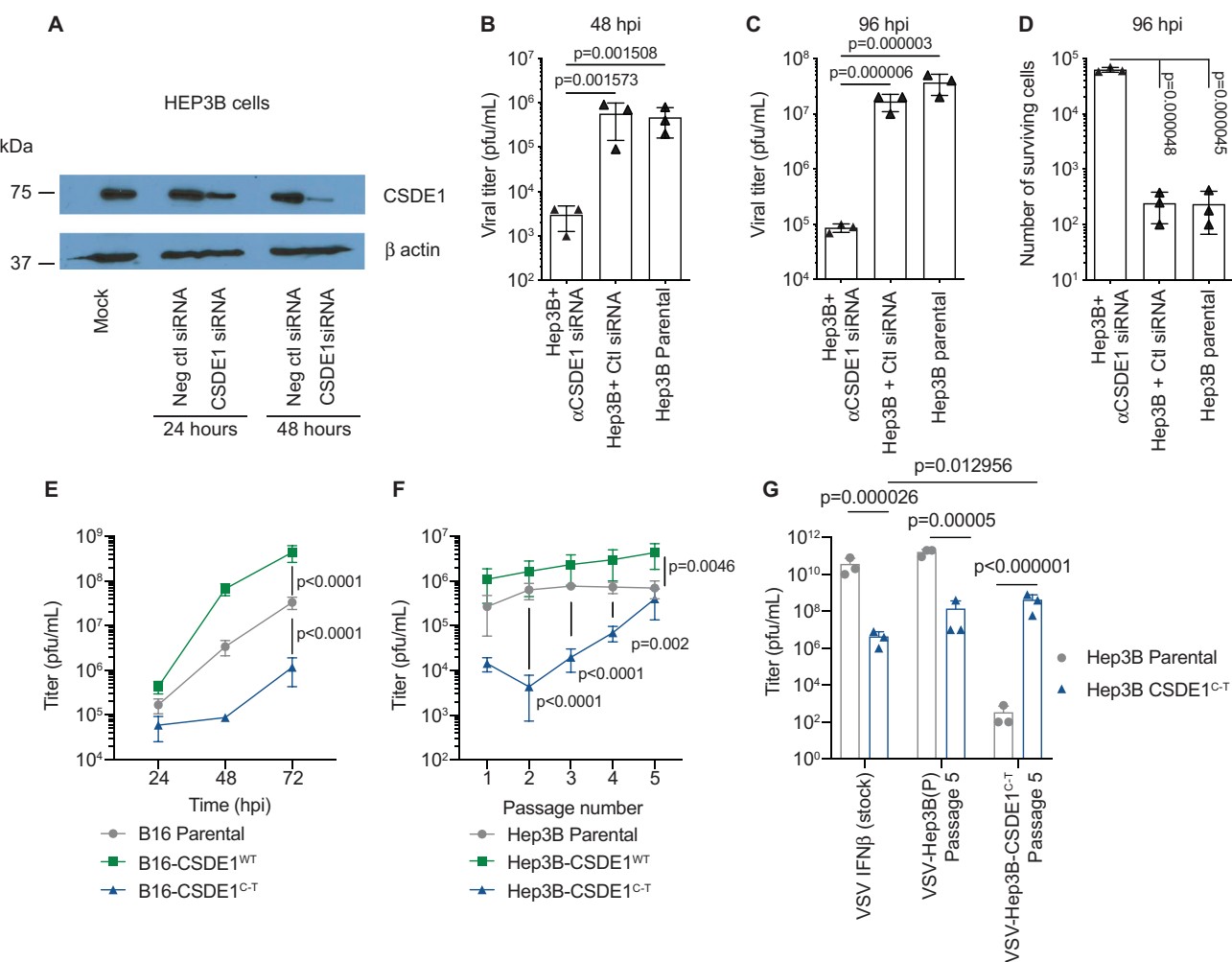

**Fig. 1 CSDE1 is a positive modulator of VSV replication. A** Hep3B cells were transfected with no siRNA, Negative control siRNA, or with [s15373 + 15374 siRNA] (2 CSDE1-specific siRNA)[46] and levels of CSDE1 assayed by western blotting 24 or 48 h later. (Representative of three separate experiments). **B–D** Forty-eight hours following transfection with siRNA as in **A**, Hep3B cells were infected with VSV-GFP (MOI 0.1). Forty-eight hours (**B**) or 96 h (**C**) later, viral titers were determined by plaque assay and **D** the number of surviving cells was counted at 96 h post infection. Representative of two separate experiments. **E** B16-, B16-CSDE1$^{C-T}$-, or B16-CSDE1$^{WT}$-overexpressing cells were infected with VSV-IFN-β at an MOI of 0.1. Twenty-four, 48, and 72 h later, viral titers were measured on BHK cells by plaque assay. Representative of three separate experiments. **F** Parental Hep3B cells or pooled populations of Hep3B-overexpressing wild-type CSDE1$^{WT}$, or mutant CSDE1$^{C-T}$, were infected with VSV-IFN-β (MOI 0.1) (3 wells/group). Forty-eight hours later (Passage 1), supernatants were assayed for infectious titers on the same cells on which the virus was passaged. Virus was recovered every 48 h (P2–5) and similarly titered. Representative of three separate experiments. **G** Stock VSV-IFN-β virus or VSV-IFN-β, which had been passaged five times through Hep3B parental or Hep3B-CSDE1$^{C-T}$ cells as in **F**, was titered on either Hep3B parental cells or on Hep3B-CSDE1$^{C-T}$ cells. Representative of two separate experiments. Means ± SD of three technical replicates are shown. *P*-values were determined using a one-way (**B–D**) or two-way (**E–G**) ANOVA with a Tukey's multiple comparisons post test on log-transformed data. Statistical significance was set at *p* < 0.05, ns > 0.05. Source data are provided as a Source Data file.

titers on Hep3B-CSDE1$^{C-T}$ cells (Fig. 1G). Conversely, virus from five passages through Hep3B-CSDE1$^{C-T}$ replicated poorly on Hep3BP cells but at near-wild-type levels on Hep3B-CSDE1$^{C-T}$ cells (Fig. 1G). Thus, VSV-IFNβ can, if given sufficient time, adapt to the emergence of escape cells by complementing the *CSDE1$^{C-T}$* mutation.

**VSV can be forced to evolve to adapt to the CSDE1$^{C-T}$ mutation.** VSV-IFNβ recovered from five passages through Hep3B-CSDE1$^{C-T}$ cells [high titer on Hep3B-CSDE1$^{C-T}$ cells, low titer on Hep3BP (Fig. 1G)] consisted of a population of quasi-species of viruses, which contained a single C–U mutation in the intergenic region (IGR) between the *P* and *M* genes (Fig. 2A) at high frequency in the whole population (Fig. 2B, C). This IGR P/M$^{C-U}$

mutation was undetectable in the stock VSV-IFNβ or in VSV-IFNβ recovered from five passages through Hep3BP (Fig. 2A–C). The same IGR P/M$^{C-U}$ mutation was recovered from VSV-IFNβ serially passaged 5× through Mel888-CSDE1$^{C-T}$ melanoma cells. Similarly, VSV-IFNβ recovered from five passages through Hep3B cells, which had previously been selected for resistance to VSV-IFNβ over 21 days (Hep3B-VSV-hIFNβ ESC), or through Mel888-21d-VSV-hIFNβ ESC cells, was almost entirely mutant for the IGR P/M$^{C-U}$ mutation (Fig. 2D). Finally, virus recovered from a Hep3B escape tumor following treatment with VSV-hIFNβ contained a mixed population of wild-type IGR P/M and mutant IGR P/M$^{C-U}$ viruses (Fig. 2E).

**CSDE1 regulates levels of viral P and M mRNA.** CSDE1, an RNA-binding protein involved in translational control[40–47],

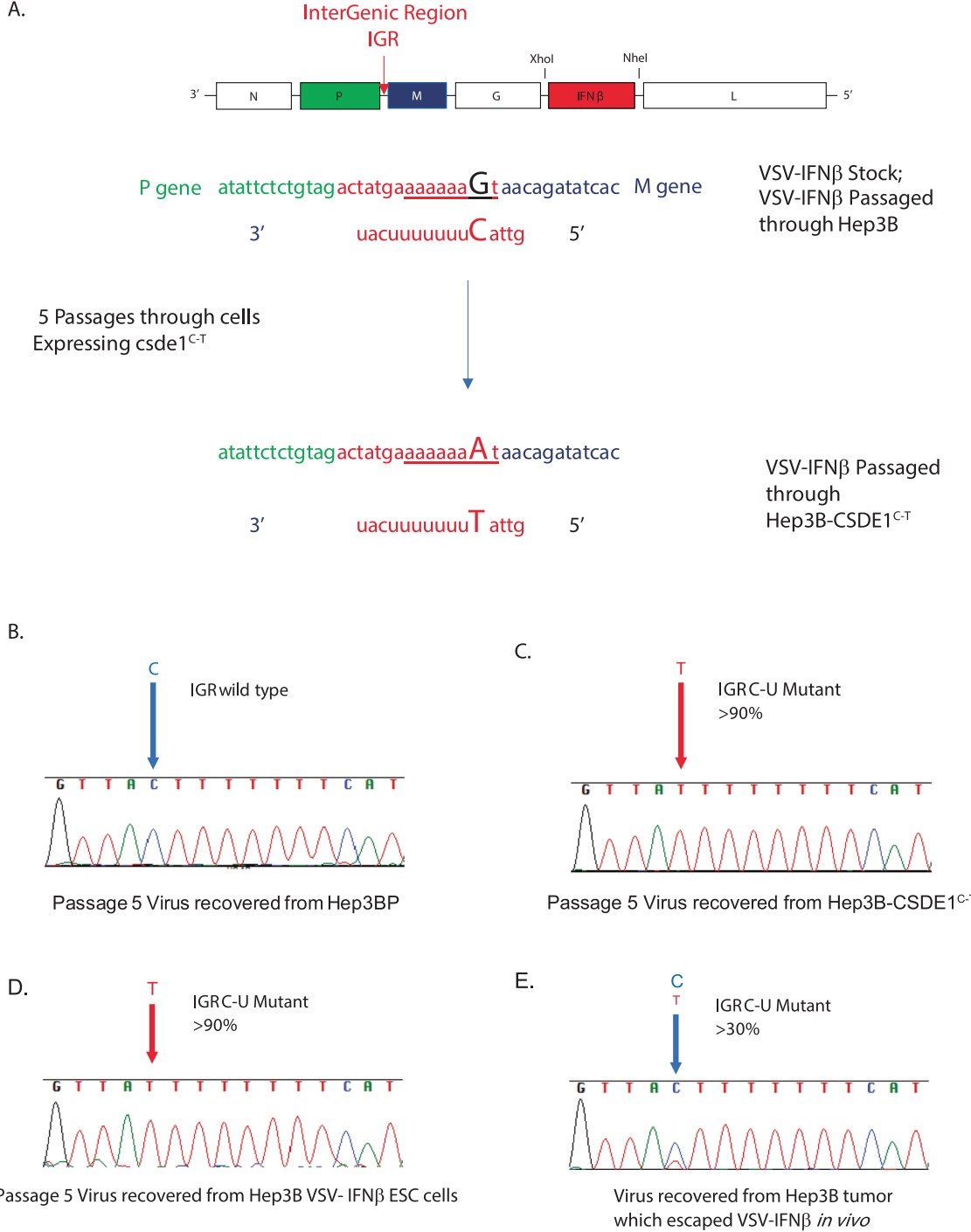

**Fig. 2 Forced evolution of VSV selects a single point mutation in the P/M IGR.** VSV-IFNβ was passaged 5× through Hep3B or Hep3B-CSDE1[C-T] cells as in Fig. 1. Sanger sequence of the IGR between the *P* and *M* genes (**A**) from virus populations passaged through **B** Hep3B parental cells showed homogenous populations of wild-type sequence; sequence of viruses passaged through either **C** Hep3B-CSDE1[C-T] cells or **D** Hep3B-VSV-IFNβ—21d ESC cells were largely homogenous for a point mutation C–U. Representative of five separate experiments. **E** Virus population from a Hep3B tumor grown in a SCID mouse and excised upon recurrence following treatment with VSV-IFNβ was a heterogenous population of wild-type and mutant IGR P/M viruses. Representative of two separate escape tumors.

binds RNA at a consensus site of 5′-(purine)(aagua)-3′[47]. The IGR P/M[C-U] point mutation C–U on the negative sense strand of the VSV genome corresponds to a G–A mutation on the positive sense strand (Fig. 2A) precisely within an exact copy of the consensus CSDE1-binding site in the IGR between the *P* and *M* genes[47] (Fig. 2A) (5′-aaaaa(aaGua)-3′ to 5′-aaaaa(aaAua)-3′). This consensus site will only be present in viral positive sense transcripts when the positive sense genomic full-length strand is made during normal replication of the VSV genome (negative to positive RNA). Alternatively, it will exist if a sub-genomic, positive strand *P–M* mRNA is made when the polymerase reads through the P/M IGR to make a unicistronic *P–M* mRNA. Such readthrough is rare, because *P* and *M* mRNAs are usually made by disengagement of

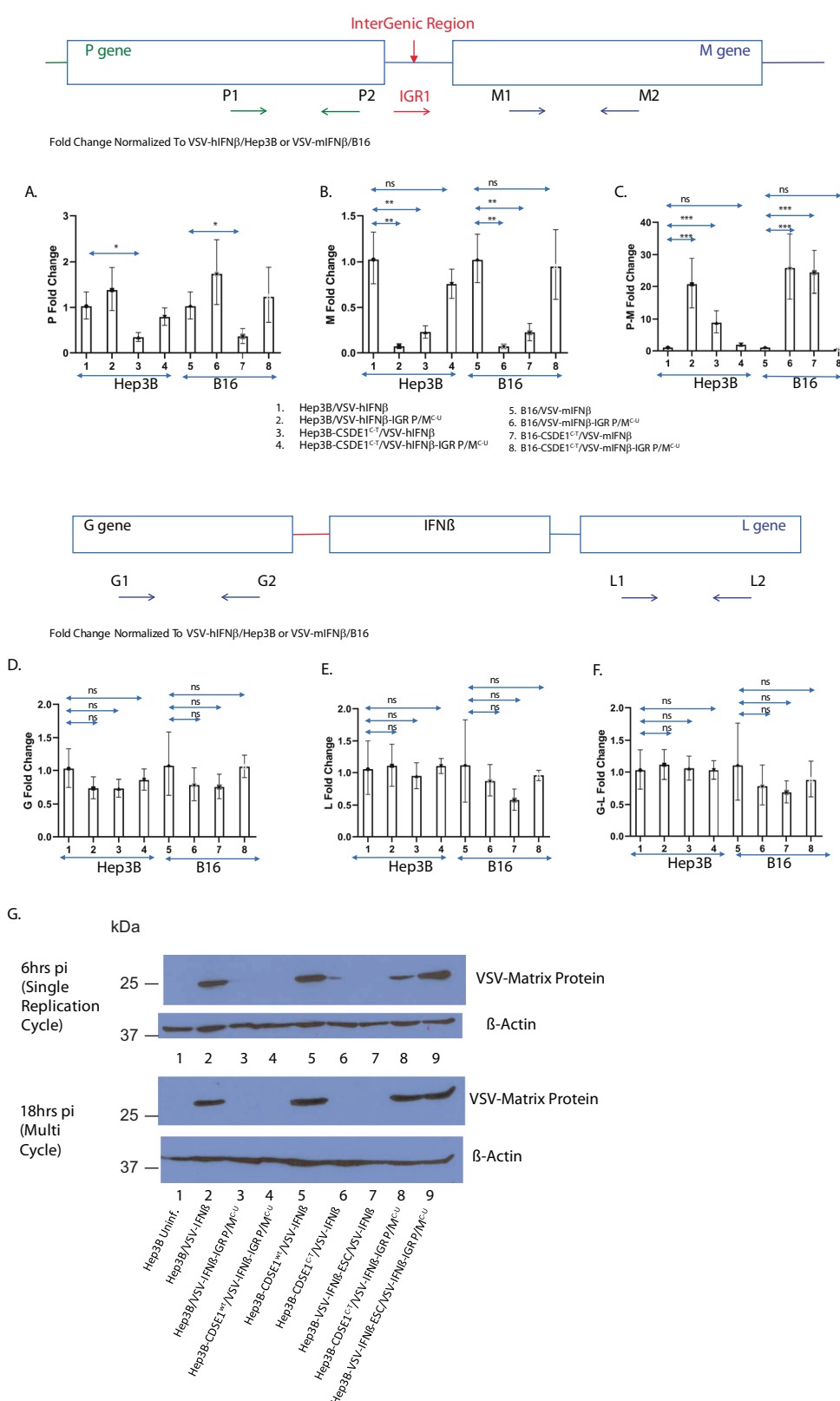

polymerase at the *P–M* IGR followed by re-initiation at the *M* gene.

Therefore, we measured *P*, *M*, and *P–M* mRNAs transcribed from VSV-IFNβ or VSV-IFNβ-IGR P/M$^{C-U}$, upon infection of B16 and Hep3BP, or B16-CSDE1$^{C-T}$ and Hep3B-CSDE1$^{C-T}$, cells within 6 h of infection, which represents an early stage of replication (Fig. 3). When levels of RNA were normalized to those observed in cells infected by the wild-type VSV-IFNβ, there were no significant changes in levels of *P* RNA in both B16 and Hep3B cells infected with VSV-IFNβ-IGR P/M$^{C-U}$ (Fig. 3A). In both Hep3B and B16 cells overexpressing CSDE1$^{P5S}$, levels of *P* RNA were slightly decreased compared to VSV-IFNβ (Fig. 3A). However,

**Fig. 3 CSDE1 regulates viral M RNA levels.** Cells (HepB3 or HepB3-CSDE1$^{C-T}$; B16 or B16-CSDE1$^{C-T}$) were infected with VSV viruses (VSV-IFNβ or VSV-IFNβ-IGR P/M$^{C-U}$) at an MOI of 3. Lanes: 1, Hep3B/VSV-hIFNβ; 2, Hep3B/VSV-hIFNβ-IGR P/M$^{C-U}$; 3, Hep3B-CSDE1$^{C-T}$/VSV-hIFNβ; 4, Hep3B-CSDE1$^{C-T}$/VSV-hIFNβ-IGR P/M$^{C-U}$; 5, B16/VSV-mIFNβ; 6, B16/VSV-mIFNβ-IGR P/M$^{C-U}$; 7, B16-CSDE1$^{C-T}$/VSV-mIFNβ; and 8, B16-CSDE1$^{C-T}$/ VSV-mIFNβ-IGR P/M$^{C-U}$. Six hours later, RNA was isolated and converted into cDNA. qrtPCR was used to assess levels of **A** viral *P* RNA (P-specific primers P1 and P2), **B** viral *M* RNA (M-specific primers M1 and M2), or **C** viral *P–M* RNA (P/M IGR-M-specific primers IGR1 and M2). Representative of three separate experiments. **D–F** Levels of *G*, *L*, and *G–L* RNA were measured as in **A–C** using primers G1 and 2, L1 and 2, and G1–L2, respectively. Representative of two separate experiments. Levels of each RNA species were normalized to expression in Hep3B infected with VSV-hIFNβ or to B16 infected with VSV-mIFNβ. **G** Hep3B parental cells (Hep3B) (Lanes 1–3) or Hep3B cells engineered to overexpress CSDE1$^{wt}$ (Hep3B-CSDE1$^{wt}$) (lanes 4 and 5) or mutated CSDE1$^{P5S}$ (Hep3B-CSDE1$^{C-T}$) (lanes 6 and 8) proteins, or Hep3B that had been selected in vitro for resistance to VSV-IFNβ oncolysis for 21 days (Hep3B-VSV-IFNβ-ESC) (lanes 7 and 9), were infected with wild-type VSV-IFNβ (lanes 2 and 5–7) or VSV-IFNβ-IGR P/M$^{C-U}$ (lanes 3, 4, 8, and 9) at an MOI of 3. Cells were collected at either 6 or 18 h post infection and levels of viral M protein were measured by western blotting. Representative of two separate experiments means ± SD of three biological replicates are shown (A–F). P-values were determined using a one-way ANOVA (A–F) on the raw data. Statistical significance was set at $p < 0.05$, ns > 0.05. *$p < 0.05$, **$p < 0.1$, ***$p < 0.001$. Source data are provided as a Source Data file.

infection of both cell types overexpressing CSDE1$^{P5S}$ with VSV-IFNβ-IGR P/M$^{C-U}$ completely normalized levels of P RNA. In contrast to moderate, or no, changes in the levels of *P* RNA, levels of *M* RNA were significantly decreased following infection of Hep3B-CSDE1$^{C-T}$ cells with VSV-IFNβ, or upon infection of Hep3BP cells with VSV-IFNβ-IGR P/M$^{C-U}$ (Fig. 3B). As for the levels of P RNA, infection of Hep3B-CSDE1$^{C-T}$ with VSV-IFNβ-IGR P/M$^{C-U}$ completely normalized the levels of *M* RNA and protein, showing that the negative effects of CSDE1$^{P5S}$ on transcription of viral *M* RNA were compensated by the presence of the IGR P/M$^{C-U}$ mutation in the viral genome (Fig. 3B). The same effects were replicated in B16 cells (Fig. 3B).

At this early stage of viral replication time point, relative to VSV-IFNβ infection of parental cell types, the levels of *P–M* bicistronic RNA were increased between 20- and 30-fold upon infection of Hep3B, or B16, cells with VSV-IFNβ-IGR P/M$^{C-U}$, and between 10- and 30-fold in both Hep3B and B16 cells overexpressing CSDE1$^{P5S}$ infected with VSV-IFNβ (Fig. 3C). Once again, infection of CSDE1$^{P5S}$-overexpressing cells with the complementing IGR P/M$^{C-U}$ compensatory mutation in VSV-IFNβ-IGR P/M$^{C-U}$ was sufficient to normalize the levels of *P–M* RNA to those levels seen in wild-type infection (Fig. 3C).

In contrast to the loss of RNA for the viral M protein, coupled with significantly increased levels of *P–M* RNA (Fig. 3B), relative levels of viral *G* and *L* RNA, as well as *G–L* RNA, were not significantly changed in Hep3B, or in B16, cells infected with VSV-IFNβ-IGR P/M$^{C-U}$, or in either cell line overexpressing CSDE1$^{P5S}$ infected with VSV-IFNβ, compared to levels in parental cells infected by VSV-IFN-β (Fig. 3D–F). These data indicate that CSDE1 specifically affects mRNA synthesis termination between P and M, and not at other IGRs of the virus. This is consistent with the IGR between the P and M genes of VSV having a unique sequence of 5′-aaaaa(aaGua)-3′—which is a perfect CSDE1 consensus binding site. In contrast, all of the remaining viral IGR (N/P, M/G, and G/L) have a similar, but distinct, sequence of 5′-aaaaa(aaCua)-3′.

These quantitative reverse transcriptase PCR (qrtPCR) data suggested that the CSDE1$^{P5S}$ protein interferes with early-stage VSV replication (at 6 h), leading to significant loss of unicistronic *M* RNA and relative increases of bicistronic *P–M* RNA. This would predict that levels of viral M protein would be significantly reduced in cells expressing mutated CSDE1$^{P5S}$ infected by VSV-IFNβ or by cells expressing wild-type CSDE1 and infected by the VSV-IFNβ-IGR P/M$^{C-U}$ virus. Consistent with this hypothesis, and with the qrtPCR results (Fig. 3B), compared to Hep3B cells infected with VSV-IFNβ (Fig. 3G, lane 2), infection with VSV-IFNβ-IGR P/M$^{C-U}$ led to markedly reduced levels of M protein at both 6 and 18 h post infection (lane 3). Overexpression of CSDE1$^{wt}$ protein in the Hep3B cells could not rescue M protein

expression following infection with VSV-IFNβ-IGR P/M$^{C-U}$ (lane 4), although it maintained, and moderately increased, the levels of M protein from infection with VSV-IFNβ (lanes 2 and 5). Overexpression of the CSDE1$^{P5S}$ mutant protein was inhibitory to replication of VSV-IFNβ as measured by the levels of M protein (lanes 2 and 6) but rescued the expression of M protein from the VSV-IFNβ-IGR P/M$^{C-U}$ virus (lane 8). In addition, Hep3B cells, which had undergone stringent in vitro selection to escape VSV-IFNβ oncolysis (Hep3B-VSV-IFNβ-ESC), and which contain the CSDE1$^{C-T}$ mutation at very high frequency (Supplementary Fig. 1), produced significantly less M protein when infected with VSV-IFNβ compared to infection with VSV-IFNβ–IGR P/M$^{C-U}$ (Fig. 3D, lanes 7 vs. 9).

**VSV expressing an escape-associated tumor antigen.** The APOBEC3B-generated *CSDE1$^{C-T}$* mutation creates a heteroclitic neo-epitope in the B16/C57Bl/6 model[11] and is selected for in tumors forced to escape VSV-IFNβ. This treatment-driven neo-antigenesis makes CSDE1$^{P5S}$ an EATA target for immunotherapy against treatment-resistant tumors. Therefore, we constructed viruses expressing either CSDE1$^{WT}$ or the CSDE1$^{P5S}$ EATA (Fig. 4A). Consistent with Fig.1, overexpression of CSDE1$^{WT}$ from VSV significantly enhanced viral replication on human and murine (but not hamster) cells, compared to VSV-GFP (Fig. 4B). Conversely, viral-driven CSDE1$^{P5S}$ exerted a significant negative effect (Fig. 4B). Low multiplicity of infection (MOI) infection of Hep3BP or B16 cells with VSV-IFNβ-CSDE1$^{WT}$ significantly reduced both escape (Fig. 4C) and the escape-enabling *CSDE1$^{C-T}$* mutation (~10–50% in Fig. 4D, compared to >90% in Supplementary Fig. 1I) compared to VSV-IFNβ.

**Trap and ambush immunotherapy for tumor escape.** Mice treated intratumorally (i.t.) with VSV-mIFNβ-CSDE1$^{WT}$ or VSV-mIFNβ-CSDE1$^{C-T}$ (Fig. 5A) generated comparable strong antiviral T-cell responses (Fig. 5B). Although VSV-mIFNβ-CSDE1$^{WT}$ did not generate α-CSDE1$^{WT}$ T cells, VSV-mIFNβ-CSDE1$^{C-T}$ induced potent T-cell responses against the CSDE1$^{P5S}$ neoantigen (Fig. 5B) as well as weaker responses against B16-CSDE1$^{WT}$ and B16 (expressing endogenous CSDE1), confirming that CSDE1$^{P5S}$ acts as a heteroclitic neo-epitope in the C57Bl/6 model[11]. These T-cell responses probably contributed to the significantly reduced tumor sizes in mice treated with VSV-mIFNβ-CSDE1$^{C-T}$ compared to those treated with VSV-mIFNβ-CSDE1$^{WT}$ at day 30 when this experiment was stopped (Fig. 5A). Only VSV-mIFNβ-CSDE1$^{C-T}$ induced intra-tumoral interleukin (IL)-12 after six i.t. injections (Fig. 5C), which correlated with the anti-tumor T-cell response (Fig. 5B). All three viruses induced similar levels of tumor necrosis factor-α (TNF-α) within injected tumors (Fig. 5D).

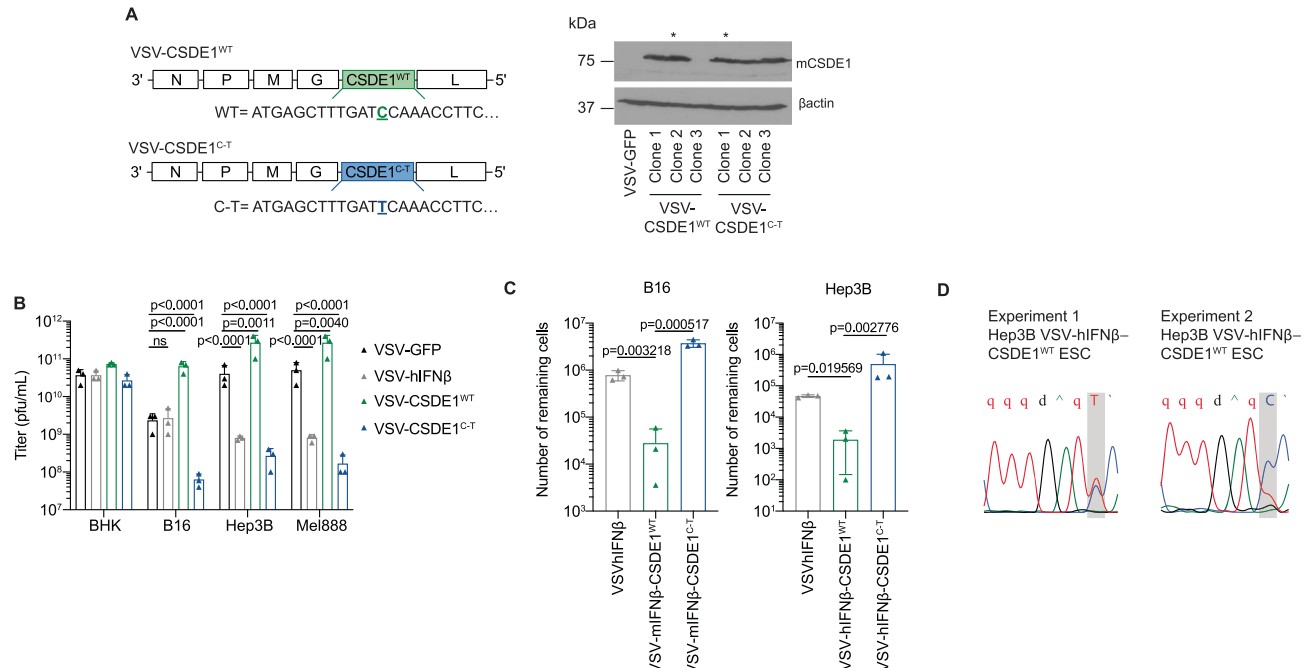

**Fig. 4 CSDE1 expressed from the virus enhances replication. A** Viruses expressing CSDE1$^{WT}$ and CSDE1$^{C-T}$ were constructed and validated by western blotting for the expression of CSDE1. Stars indicate clones that were used in subsequent experiments. Representative of three separate experiments. **B** BHK (hamster), B16 (mouse), Hep3B, and Mel888 (human) cell lines were infected with VSV-GFP, VSV-hIFN-β, VSV-CSDE1$^{WT}$, or VSV-CSDE1$^{C-T}$ at an MOI of 3 (triplicate wells per cell line). Forty-eight hours later, virus was titered on BHK cell by plaque assay. Representative of three separate experiments. **C** Murine B16 or human Hep3B cells were infected (MOI 0.01) with VSV-IFNβ, VSV-IFN-β-CSDE1$^{WT}$, or with VSV-IFN-β-CSDE1$^{C-T}$ viruses (using species-matched IFNβ genes) for 21d as in "Methods" and ref. [10]. Surviving cells were pooled and counted. **D** Hep3B cells were infected (MOI 0.01) with the VSV-hIFNβ-CSDE1$^{WT}$ virus for 21d as in "Methods" and ref. [10]. Surviving cells were pooled and genomic DNA prepared. Sanger sequencing of CSDE1 is shown for two independent experiments. In these experiments, as well as in two other experiments, a mixed population of mutated and unmutated cells were selected. Means ± SD of three technical replicates are shown. *P*-values were determined using a two-way (**B**) or one-way (**C**) ANOVA with a Tukey's multiple comparisons post test on log-transformed data. Statistical significance was set at *p* < 0.05, ns > 0.05. Source data are provided as a Source Data file.

Immune checkpoint blockade (ICB) with α-PD-1 antibody[49–52] concomitant with i.t. virus, significantly decreased IL-12 in VSV-mIFNβ-CSDE1$^{C-T}$-treated tumors (Fig. 5C). In contrast, αPD-1 ICB 4 days after the first viral injection significantly increased IL-12 in VSV-mIFNβ-CSDE1$^{C-T}$-treated tumors (Fig. 5C). Levels of TNF-α were not significantly altered from no, or early, ICB (Fig. 5D).

To compare the relative therapeutic contributions of increased viral replication/oncolysis (VSV-mIFNβ-CSDE1$^{WT}$) with decreased oncolysis but treatment-driven neo-antigenesis in VSV-IFNβ-ESC tumors (VSV-mIFNβ-CSDE1$^{C-T}$), mice were treated i.t. with viruses + α-PD-1 late after induction of T-cell responses (Fig. 5E). VSV-mIFNβ prolonged survival compared to phosphate-buffered saline (PBS), but all tumors eventually escaped (Fig. 5F). VSV-IFNβ-CSDE1$^{WT}$ significantly increased median survival compared to VSV-mIFNβ (Fig. 3F), correlated with enhanced i.t. replication (Fig. 5G) (consistent with Fig. 4). However, expression of the CSDE1$^{P5S}$ EATA from the virus completely prevented tumor escape (Fig. 5F), despite significantly less replication in tumors compared to either VSV-mIFNβ or VSV-mIFNβ-CSDE1$^{WT}$ (Figs. 5G and 4). It is unlikely that evolution of the escape-promoting CSDE1$^{P5S}$ mutation occurs in 100% of all cells in the ESC tumors (e.g., see Supplementary Fig. 1J). Therefore, our model of tumor clearance depends upon the heteroclitic anti-CSDE1$^{P5S}$/anti-CSDE1$^{WT}$ T-cell response being potent enough to clear that proportion of tumor cells in which the CSDE1$^{P5S}$ mutation had not evolved following VSV-IFNβ therapy. In this respect, as we have seen previously,

adoptive transfer of in vitro-activated OT-I CD8+ T cells (specific for the irrelevant SIINFEKL epitope of Ovalbumin) in combination with anti-PD-1 ICB had no significant therapeutic effect upon the growth of subcutaneous B16 tumors (100% CSDE1$^{WT}$) (Supplementary Fig. 3A). In contrast, CD8+ T cells recovered from mice that had survived B16 tumors treated with VSV-CSDE1$^{P5S}$ (Fig. 5F) and expanded in vitro against the mutated CSDE1$^{P5S}$ MFSDSNLLH peptide, significantly extended survival compared to the control-treated group, and cured a proportion of mice (Supplementary Fig. 3A). Addition of ICB with anti-PD-1 antibody significantly further enhanced the efficacy of the adoptive transfer of anti-CSDE1$^{P5S}$ CD8+ T cells and cured 100% of mice (Supplementary Fig. 3A). Finally, adoptive transfer of anti-CSDE1$^{P5S}$ CD8+ T cells in combination with frontline treatment with VSV-IFNβ also cured all the mice (even in the absence of ICB), whereas a combination of VSV-IFNβ and OT-1 CD8+ T cells was no more effective than the virus alone (Supplementary Fig. 3B). These data show that T cells raised against mutant CSDE1$^{P5S}$ are therapeutically sufficient to treat tumors expressing CSDE1$^{WT}$ antigen, despite the weaker strength of the heteroclitic response against B16 cells compared to that against B16-CSDE1$^{P5S}$-expressing ESC cells.

**Dendritic cell vaccination against an escape-associated tumor antigen.** To separate the conflicting effects of decreased oncolysis (Figs. 4 and 5G) against neo-antigenesis (Fig. 5F), we tested EATA-targeted immunotherapy using dendritic cells (DC)

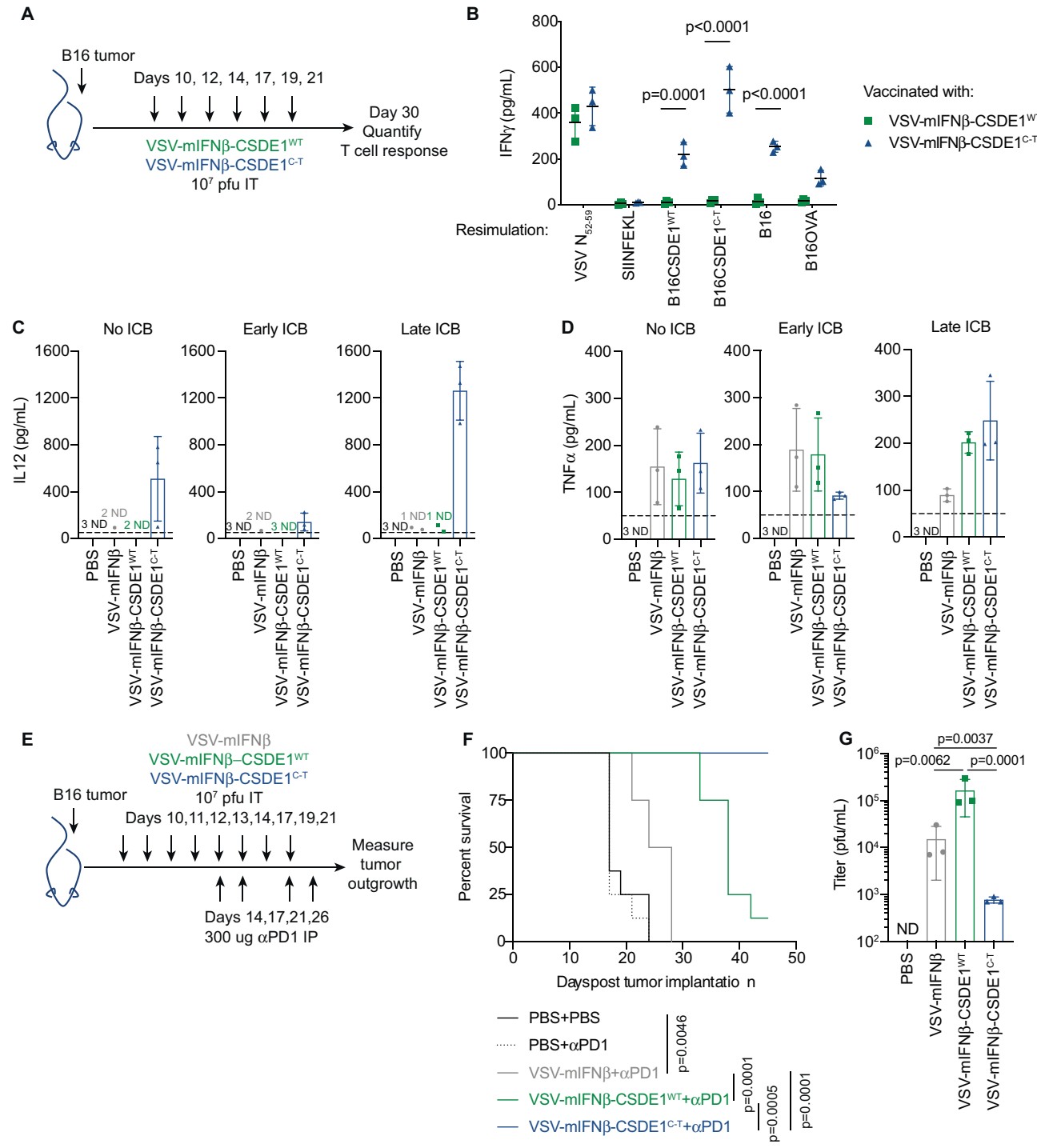

expressing CSDE1$^{P5S}$. DC-CSDE1$^{C-T}$ also generated strong α-CSDE1$^{P5S}$ T-cell responses (Supplementary Fig. 2A, B). VSV-IFNβ + DC-CSDE$^{C-T}$ + α-PD-1 significantly enhanced therapy relative to VSV-IFNβ + DC-CSDE$^{WT}$ + α-PD-1 (Supplementary Fig. 2C), but never achieved the 100% cure rates of VSV-IFNβ-CSDE1$^{C-T}$ + α-PD-1 (Fig. 5F), which correlated with ~3-fold lower levels of i.t. IL-12 (Supplementary Fig. 2D and Fig. 5C).

**Human tumor cells that escape VSV-IFNβ are immunogenic.** We investigated whether CSDE1$^{P5S}$, and other undefined EATA, would be immunogenic for human T cells. In three separate co-cultures, CD3/CD28-activated human CD3+ T cells

had different baseline reactivity against Hep3B targets, reflecting different alloreactivities[11,53–55] (Fig. 6A, B). Nonetheless, both APOBEC3B-mutated Hep3B targets and *CSDE1$^{C-T}$*-expressing Hep3B-VSV-hIFNβ-ESC cells (Supplementary Fig. 1I) were significantly more immunogenic than Hep3BP across all three donors (Fig. 6B). Immunogenicity was significantly reduced by knockdown of CSDE1[46] (Fig. 6B), suggesting that neo-antigenesis of hCSDE1$^{P5S}$ may serve as an EATA. T cells expanded against Hep3B were unable to kill either Hep3B or Hep3B-VSV-IFNβ-ESC targets (Fig. 6C). In contrast, T cells expanded against Hep3B-VSV-IFNβ-ESC cells showed significant cytotoxicity against Hep3B-VSV-IFNβ-ESC targets (Fig. 6C) and some

**Fig. 5 Virotherapy trap and immunotherapy ambush. A** C57Bl/6 mice bearing 10d B16 tumors were injected i.t. with PBS, VSV-mIFNβ, VSV-mIFNβ-CSDE1$^{WT}$, or VSV-mIFNβ-CSDE1$^{C-T}$. At day 30, mice were killed for collecting splenocytes and tumor sizes were measured as shown. **B** Splenocytes collected at d30 were re-stimulated with VSV-N-specific immunodominant peptide (N$_{52-59}$), SIINFEKL peptide from OVA, or with B16 cells overexpressing CSDE1$^{WT}$ or CSDE1$^{C-T}$, or with B16 or B16ova cells (E : T 10 : 1) for 72 h. Supernatants were assayed for IFN-γ. Representative of two separate experiments. **C**, **D** Mice were injected with viruses as in **A** with added groups (three mice per group), which received no immune checkpoint blockade (ICB), or αPD-1 antibody i.p. at days 10, 12, and 14 or at days 14, 17, and 21. Twenty-four hours after the last injection of virus (d22), or earlier if tumor > 1.0 cm diameter (PBS groups), tumors were dissociated and assayed for **C** IL-12 or **D** TNF-α by ELISA (normalized by protein concentration in whole tumor lysates as pg/ml protein). **E** C57Bl/6 mice with 10d B16 tumors were injected i.t. with PBS, VSV-mIFNβ, VSV-mIFNβ-CSDE1$^{WT}$, or with VSV-mIFNβ-CSDE1$^{C-T}$ (10$^7$ pfu/injection) followed by α-PD-1 (n = 8/group). **F** Kaplan–Meier survival for groups in **E**. P-values were determined using the log-rank Mantel–Cox test. For multiple comparisons using the Bonferroni correction, overall statistical significance threshold was set at $\alpha = 0.05$ (3 comparisons at $p < 0.0125$ (4 comparisons)). Representative of two separate experiments. **G** C57Bl/6 mice with 10d B16 tumors were injected i.t. with (column 1) PBS, (2) VSV-mIFNβ, (3) VSV-mIFNβ-CSDE1$^{WT}$, or (4) VSV-mIFNβ-CSDE1$^{C-T}$ (10$^7$ pfu/injection) on days 10, 11, and 12. On d13, tumors were excised and virus measured by plaque assay on BHK cells. Representative of two separate experiments. Each symbol in **B–D** and **G** represents a mouse (n = 3/group). Means ± SD are shown. ND, not detected (below limit of detection). P-values were determined using a two-way (**B**) or one-way ANOVA (**C**, **D**, and **G**) with a Tukey's multiple comparisons post test. Statistical testing was performed on log-transformed data in **G**. Statistical significance set at $p < 0.05$ for **B–D** and **G**. Source data are provided as a Source Data file.

cytotoxicity against Hep3B, suggesting that some T-cell responses against EATA may be heteroclitic. Of all possible 8-, 9-, 10-, and 11-mers with the Proline–Serine mutation at amino acid 5, the 9mer, MSFDSNLLH, was predicted to have weak binding affinity for human leukocyte antigen- (HLA) A*01:01, HLA-A*03:01, and HLA-B*58:01 (Fig. 6D), which, for HLA-A*01:01 and HLA-B*58:01, was predicted to be stronger than the wild-type epitope MSFDPNLLH. T cells from one additional donor could be primed by CSDE1$^{C-T}$-transfected DC, but not by CSDE1$^{WT}$-transfected DC, to recognize the CSDE1$^{P5S}$ EATA. However, T cells from a second donor did not recognize either wild-type or mutated CSDE1 (Fig. 6E). Both donors 4 and 5 showed high level T-cell priming against Hep3B-VSV-IFNβ-ESC cells compared to Hep3B (Fig. 6E), as with Fig. 6B. Thus, escape from VSV-hIFNβ generated cells that were consistently more immunogenic than parental in both human and murine contexts.

CD3+ T cells from one of three additional donors secreted significantly more IFN-γ when stimulated in vitro with DC-presented, mutated CSDE1$^{P5S}$ 9mer peptide (MSFDSNLLH) compared to either the DC-presented wild-type CSDE1 9mer (MSFDPNLLH) or the negative control SIINFEKL peptide (Fig. 6F). However, CD3+ T cells from all three donors secreted significantly more IFN-γ when stimulated in vitro with a 20mer peptide in which the Pro-Ser mutation in CSDE1$^{P5S}$ could potentially be presented by DC at any position in a loaded HLA molecule, compared to when DC were loaded with the wild-type 20mer or the SIINFEKL peptide (Fig. 6F). These data provide additional support for the hypothesis that neo-antigenesis of hCSDE1$^{P5S}$ serves as an EATA.

## Discussion

Here we exploited neo-antigenesis resulting from high mutational plasticity of tumors, which also facilitates treatment escape, to impose a powerful immunotherapy against escape tumors as they are forced to evolve in response to frontline treatment. By targeting a predictable and reproducible mutation induced with high clonality within treatment-escape tumors, we were able to improve the efficacy of VSV-IFNβ viro-immunotherapy significantly over that obtained with the virotherapy alone.

Escape from treatments such as oncolytic virotherapy can occur for multiple reasons[1–4], involving not only tumor cell mutational plasticity but also other mechanisms including a simple lack of efficient infection, HLA incompatibility with EATA, immune suppression, and antiviral tumor microenvironments. However, we show that mutational pathways, such as APOBEC3B, induced by frontline treatment with our clinical agent VSV-IFNβ lead to the emergence of escape variants

carrying a very specific mutation, which is heavily selected for at high frequency (Supplementary Fig. 1). We reasoned that such mutations may be in genes/proteins that mediate escape from innate and adaptive immune-mediated mechanisms of tumor clearance induced by VSV infection[26–31], and/or may allow infected cells to downregulate critical steps in viral replication and thereby escape oncolysis.

CSDE1 is multi-functional RNA-binding protein that regulates RNA translation[40–47]. CSDE1 has not previously been reported to be involved in the regulation of VSV replication, although it has been shown to stimulate cap-independent translation initiation for several other viruses (reviewed in ref. [45]). Thus, knockdown of CSDE1 reduced the internal ribosome entry site (IRES)-driven translation of both human rhinovirus (HRV) and poliovirus, while not affecting cap-dependent translation[56]. In the case of HRV-2, CSDE1 binding to viral mRNA alters its structure to facilitate further binding of the polypyrimidine tract-binding protein, creating a structure that is necessary for translational competency[57]. Thus, CSDE1 can be viewed as an RNA chaperone, which facilitates the formation of tertiary protein/RNA complexes, thereby bridging viral RNAs and proteins that cannot bind directly to each other.

Across species and tumors, knockdown of CSDE1 significantly decreased VSV replication, whereas its overexpression enhanced virus replication (Figs. 1 and 4). Overexpression of CSDE1$^{P5S}$ also significantly decreased VSV replication (Figs. 1 and 4), despite intact endogenous CSDE1$^{WT}$ protein. Therefore, this point mutation creates a mutant protein, which exerts a potent inhibitory effect on VSV replication.

VSV forced to evolve on cells overexpressing CSDE1$^{P5S}$ initially had very low titers, but adapted to recover its fitness after multiple passages (Fig. 1), which correlated with the emergence of a compensatory C–U mutation in the virus at the P/M IGR (Fig. 2). Thus, if given sufficient time, VSV-IFNβ was able to adapt to treatment-escape cells, although viral adaptation lagged behind tumor mutation/evolution, thereby allowing treatment escape.

The data in Fig. 3 show that mutant CSDE1$^{P5S}$ acts at the P/M IGR of wild-type VSV-IFNβ, leading to significant reductions in levels of $M$ mRNA and protein, as well as increased levels of aberrant bicistronic $P$–$M$ RNA. Virus carrying the IGR P/M$^{C-U}$ mutation in the presence of only wild-type CSDE1$^{WT}$ generated similarly aberrant levels of $M$ mRNA and protein. However, the combination of cellular CSDE1$^{P5S}$ protein and viral IGR P/M$^{C-U}$ mutation act in a self-complimentary manner to normalize viral $P$ and $M$ RNA levels for efficient viral replication. These data are consistent with the presence of a predicted consensus binding site for CSDE1[47] —5′-aaaaa(aaGua)-3′—on the positive sense

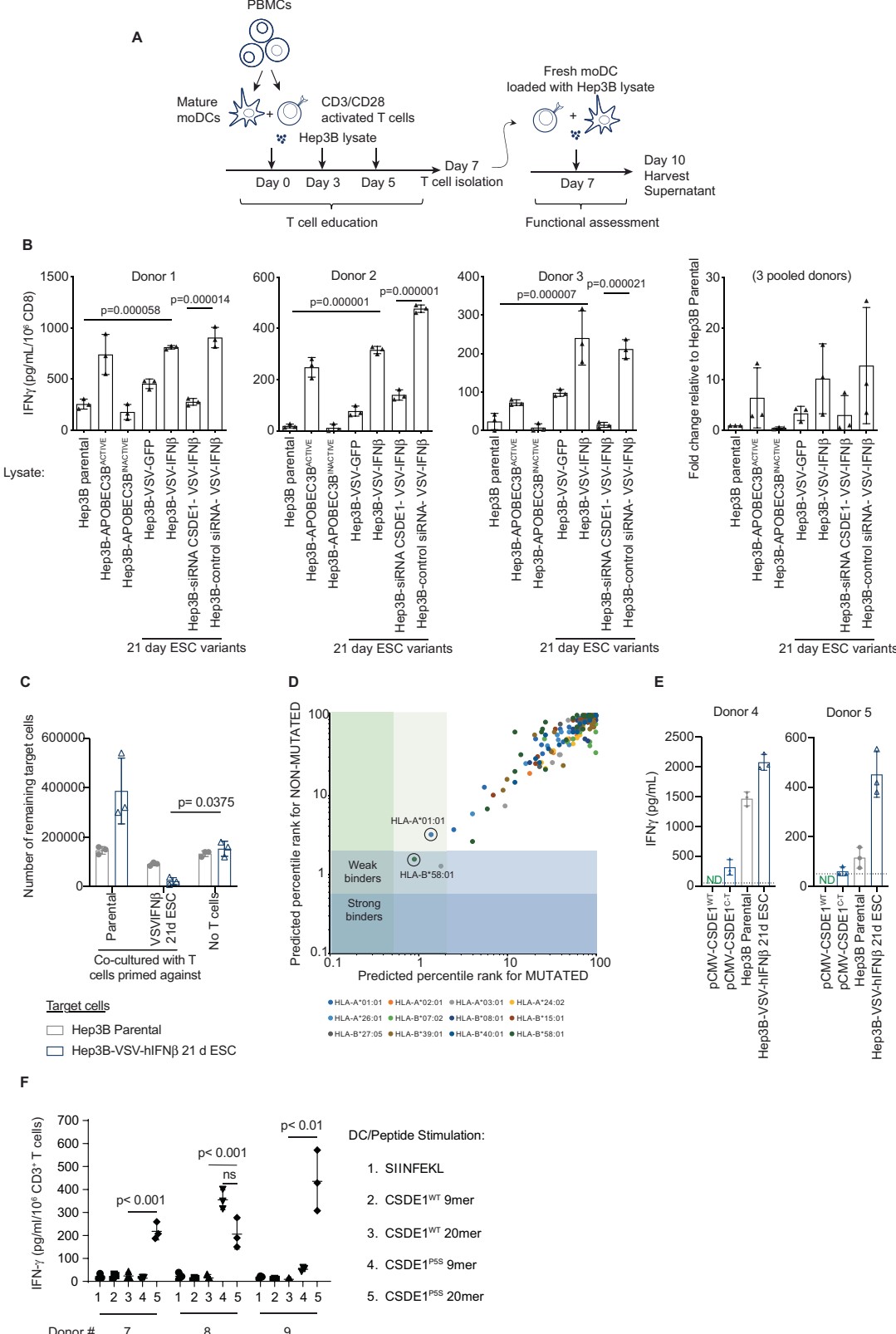

genomic strand of the VSV genome in the IGR between the *P* and *M* genes (Fig. 2A). Overall, these data suggest a model in which CSDE1[WT] binding to the consensus binding site on the positive sense strand at the P/M IGR facilitates viral polymerase disengagement to make unicistronic *P* and *M* mRNAs. Mutated CSDE1[P5S] may inhibit this disengagement, thereby significantly deregulating viral transcription, leading to decreased unicistronic *M* RNA (Fig. 3B), increased bicistronic *P–M* RNA (Fig. 3C), and loss of M protein (Fig. 3G). In this scenario, the complementary mutation of the viral IGR P/M[C-U] would allow mutated

**Fig. 6 Escape from VSV-hIFNβ is immunogenic. A** Human CD3+ T cells activated with αCD3 and αCD28 were co-cultured with autologous CD14+ -matured DC and different Hep3B cell lysates on d1, 3, and 5. On d7, CD8+ T cells were isolated and co-cultured with autologous DC and the same Hep3B cell lysates (E : T 10 : 1). **B** Seventy-two hours later, supernatants were assayed for IFN-γ. Means ± SD of three technical replicates from three donors. **C** $10^4$ target cells (Hep3BP or Hep3B-VSV-hIFN-β 21d ESC) were treated for 24 h with hIFN-γ before being co-cultured with $10^5$ T cells primed/expanded on Hep3BP or Hep3B-VSV-hIFNβ-21d ESC cells as in **A**. $10^5$ T cells were added after 48 h. At 120 h, surviving adherent cells were counted. **D** NetMHC4.0 % rank of the predicted affinity of the unmutated CSDE1$^{WT}$ 9mer, MSFDPNLLH, and its CSDE1$^{C-T}$-mutated counterpart 9mer, MSFDSNLLH, compared to 400,000 random peptides for HLA subtypes. Strong binders, %rank < 0.5, weak binders, %rank < 2. http://www.cbs.dtu.dk/services/NetMHC/. **E** Human CD3+ T cells activated in vitro were co-cultured with autologous DC and Hep3BP or Hep3B-VSV-hIFNβ-ESC lysates, or with DC transfected 48 h previously with 10 μg pcDNA3.1-CSDE1$^{WT}$ or pcDNA3.1-CSDE1$^{C-T}$ plasmids. Lysates, or DC, were re-added on d5. On d7, isolated CD8+ T cells were co-cultured with DC/Hep3B cell lysates or with transfected DC (E : T 10 : 1). Seventy-two hours later, supernatants were assayed for IFN-γ. Means ± SD of three technical replicates from two donors. ND, below limit of detection. **F** Activated human CD3+ T cells were co-cultured with autologous DC pulsed 48 h previously with 5 μg/ml peptide on d1 and 5. On d7, isolated CD8+ T cells were co-cultured with similarly pulsed DC. Seventy-two hours later, supernatants were assayed for IFN-γ. Peptide pulsing: Lane 1: SIINFEKL peptide from Ovalbumin; 2: WT CSDE1 9mer MSFDPNLLH; 3: CSDE1 20mer, MSFDPNLLHNNGHNGYPNGT, which could be processed into 9-11mer with Pro (position 5) in different positions; 4: mutated CSDE1$^{P5S}$ MSFDSNLLH; or 5: CSDE1$^{P5S}$ 20mer, MSFDSNLLHNNGHNGYPNGT, in which Ser (position 5) could be at different positions. Means ± SD of three technical replicates from three donors. *P*-values determined using a one-way (**B**) or two- way (**C**, **F**) ANOVA with a Tukey's multiple comparisons post test. Statistical testing was on log-transformed data in **C**. Statistical significance, *p* < 0.05, ns > 0.05. Source data are provided as a Source Data file.

CSDE1$^{P5S}$ to bind to the mutated consensus site and restore wild-type relative levels of *P*, *M*, and *P–M* readthrough RNA. Our data suggest that CSDE1 specifically affects mRNA synthesis termination between P and M, but not between other viral genes, despite the IGR sequences of VSV being very similar. However, the IGR sequence between P and M (5′-aaaaa(aaGua)-3′) differs from all the other IGR by a single base. Thus, the IGR sequence between N and P, M and G, and G and L is (5′-aaaaa(aaCua)-3′). This C–G change, which is only present in the P/M IGR, converts the sequence (5′-aaaaa(aaCua)-3′) into a perfect CSDE1 consensus binding site (5′-aaaaa(aaGua)-3′). Therefore, we hypothesize that CSDE1 specifically affects mRNA synthesis termination between P and M, and not at other very similar IGRs of the virus, because the P/M IGR is the only site with the perfect CSDE1 consensus binding site. Experiments are underway to determine whether CSDE1 binds only at the perfect CSDE1 site in the P/M IGR (and not at all at the other viral IGRs) or whether there are different levels of CSDE1 binding throughout the viral IGRs.

Taken together, these data show (i) that CSDE1 is a major positive regulator of VSV replication; (ii) that CSDE1$^{P5S}$ acts as an inhibitor of VSV replication facilitating escape from viral oncolysis; (iii) and that, although tumor cells readily evolve to escape viral therapy (CSDE1$^{WT}$ to CSDE1$^{P5S}$), the oncolytic virus can, if given sufficient time, also itself evolve to complement those mutations that occur in its replication substrate.

Consistent with this model, CSDE1 expressed from VSV (VSV-IFNβ-CSDE1$^{WT}$) enhanced replication in vitro and in vivo (Figs. 4B and 5G), reduced escape (Fig. 4C), inhibited evolution of the escape-promoting *CSDE1*$^{C-T}$ mutation (Fig. 4D), and was significantly more effective than our current clinical agent VSV-IFNβ (Fig. 5E, F). These results support the development of VSV-IFNβ-CSDE1 as a novel, improved clinical candidate beyond VSV-IFNβ.

We developed VSV expressing IFNβ to increase antiviral safety and anti-tumor immunogenicity[25,32]. However, addition of IFNβ unexpectedly increased escape through increased APOBEC3B[10], resulting in enhanced clonality of *CSDE1*$^{C-T}$ compared to VSV-GFP-ESC cells (Supplementary Fig. 1). Although this was an unexpected byproduct of inclusion of IFNβ into the virus, expression of the selected *CSDE1*$^{C-T}$ mutation in escape cells presented an opportunity, which we sought to exploit through targeting of this escape-induced mutation. Thus, as *CSDE1*$^{C-T}$ encodes a heteroclitic neo-epitope in the C57Bl/6 model[11] (Fig. 5B), we reasoned that, by forcing evolution of tumors to express *CSDE1*$^{C-T}$ through virotherapy (neo-antigenesis), escape

variants could be ambushed by T-cell responses against this predictable CSDE1$^{P5S}$ EATA. As VSV is an excellent platform for vaccination against tumor antigens[26,33,58–63], we co-expressed *CSDE1*$^{C-T}$ from VSV-IFNβ to prime escape-specific T-cell responses. VSV-IFNβ-CSDE1$^{C-T}$ replicated significantly less well than VSV-IFNβ or VSV-IFNβ-CSDE1$^{WT}$ (Figs. 4B and 5G), but induced potent T-cell responses against the CSDE1$^{P5S}$ EATA (Fig. 5B), which completely prevented escape (Fig. 5F) in the presence of anti-PD-1 ICB. Anti-CSDE1$^{P5S}$ T cells still had anti-tumor efficacy without ICB as evidenced by the significant reduction in tumor volumes in Fig. 5A. In addition, anti-CSDE1$^{P5S}$ CD8 T cells had anti-tumor activity against B16 tumors even in the absence of anti-PD-1 ICB (e.g., Supplementary Fig. 3A) and were active against B16 tumors, which escaped VSV-IFNβ therapy without anti-PD-1 ICB, at least in the context of in vivo activation and adoptive T-cell transfer (Supplementary Fig. 3B). Although VSV-mIFNβ-CSDE1$^{WT}$ was a significantly better oncolytic than VSV-IFNβ (Figs. 4B, C and 5F, G), it did not generate α-CSDE1$^{WT}$ or α-CSDE1$^{P5S}$ T-cell responses (Fig. 5B), suppressed evolution of the CSDE1$^{P5S}$ immunogen in escaping cells (Fig. 4D), and was not as effective as VSV-IFNβ-CSDE1$^{C-T}$ (Fig. 5F). Thus, the therapeutic value of T-cell control of emerging escape variants outweighed the loss of oncolytic potency of VSV-IFNβ-CSDE1$^{C-T}$ (Fig. 5F).

VSV-IFNβ-ESC tumors in vivo rarely contained a completely homogenous population of *CSDE1*$^{C-T}$ mutant tumor cells (Supplementary Fig. 1J). Therefore, the heteroclitic anti-CSDE1$^{P5S}$ T-cell responses[11] (Fig. 5B) probably contribute a significant bystander effect against tumor cells, which do not become infected, escape direct oncolysis, or innate immune clearance, or which do not evolve the *CSDE1*$^{C-T}$ mutation. This model is supported by the data in Supplementary Fig. 3, in which adoptive transfer of anti-CSDE1$^{P5S}$ CD8+ T cells improved survival of mice bearing B16 (CSDE1$^{WT}$) tumors (Supplementary Fig. 3A) or cured them when used in combination with either anti-PD-1 ICB (Supplementary Fig. 3A) or with frontline, CSDE1$^{P5S}$-inducing VSV-IFNβ therapy (without ICB) (Supplementary Fig. 3B).

Thus, it may be that intra-tumoral IL-12 (Fig. 5C), which correlated with anti-CSDE1$^{P5S}$ T-cell responses (Fig. 5B), reflects T-cell killing of both CSDE1$^{P5S}$-positive tumor cells and heteroclitic T-cell responses against CSDE1$^{WT}$ cells (Fig. 5F). DC-CSDE1$^{C-T}$, with intra-tumoral VSV-IFNβ + αPD-1, was not as effective as when the neoantigen was expressed from within the virus (Supplementary Fig. 2C) and was associated with lower intra-tumoral IL-12 (Fig. 5C and Supplementary Fig. 2C). These data are consistent with a model in which intra-tumoral

VSV-IFNβ-CSDE1$^{C-T}$ provides both high levels of inflammation (TNF-α in all VSV-injected tumors; Fig. 5D and Supplementary Fig. 2D), to enhance trafficking of anti−CSDE1$^{P5S}$-specific T cells. Simultaneously, VSV-IFNβ-CSDE1$^{C-T}$ also provides high concentrations of target antigen (CSDE1$^{P5S}$) (reflected by IL-12 only in VSV-IFN-β-CSDE1$^{C-T}$-injected tumors; Fig. 5C), which are lacking with intraperitoneal (i.p.) DC and intra-tumoral VSV-IFNβ.

We targeted the CSDE1$^{P5S}$ mutation in cells, which escaped VSV-IFNβ therapy, because it was the highest frequency mutation in VSV-IFNβ ESC cells and, therefore, represents a "trunk-like" mutation in cells escaping VSV-IFNβ (Supplementary Fig. 1). However, applying intense immunotherapeutic pressure against the CSDE1$^{P5S}$ mutation, as in Fig. 5, may allow other, lower frequency (branch-like) escape-induced mutations to become more prominent in ESC populations, to compensate for a requirement of ESC tumors to lose detectable expression of CSDE1$^{P5S}$ completely. These mutations may be in cellular proteins/pathways affecting, e.g., viral replication, the antiviral response and/or antigen presentation.

Human VSV-hIFNβ-ESC tumor cells were also significantly more immunogenic than untreated cells (Fig. 6), implying neo-antigenesis of EATA. These could include CSDE1$^{P5S}$, which was present at high clonality (Supplementary Fig. 1I), knockdown of which significantly reduced T-cell activation (Fig. 6B). Translation of virotherapy with escape-targeting immunotherapy will require identification of HLA-/patient-specific EATA, such as CSDE1$^{P5S}$ where HLA compatibility is predicted (Fig. 6D), or the simultaneous targeting of multiple (unidentified) EATA—e.g., using VSV-expressed cDNA libraries derived from treatment-escape tumors[58–60].

In summary, we have exploited the inherent genetic plasticity of tumors by using oncolytic virotherapy to drive them into an escape phenotype, which can then be ambushed by vaccination against a predictably arising EATA. This approach is likely to be widely applicable across a range of different frontline therapies, which are potent enough to drive tumor cell mutation/evolution, thereby inducing neo-antigenesis resulting in a novel immuno-peptidome associated with acquired treatment resistance.

## Methods

**Experimental design**. These experiments were designed to evaluate how reproducible mutations induced in tumor cells escaping oncolytic virotherapy could be exploited for the design of immunotherapies targeting treatment escape. The investigators were not blinded to the allocation of groups during experiments or subsequently during the analysis. Although statistical methods were not used to predetermine the sample size, sample sizes were chosen on the basis of estimates from pilot experiments and previously published results. Seven to ten mice per group were used for each survival experiment to achieve statistical power, to make multiple comparisons. Animals were randomized to treatment groups following tumor implantation using the GraphPad QuickCalcs online tool (https://www.graphpad.com/quickcalcs/randMenu/). The *n*-values and particular statistical methods are indicated in the figure legends and the statistical analysis section.

**Cell lines and viruses**. B16 murine melanoma and human Hep3B hepatocellular carcinoma and BHK cells were originally obtained from American Type Culture Collection (ATCC). Human Mel888 melanoma cells were obtained from the Imperial Cancer Research Fund in 1997/1998 and were grown in Dulbecco's modified Eagle medium (DMEM; Hyclone, Logan, UT, USA) + 10% fetal bovine serum (FBS) (Life Technologies). Cell lines were authenticated by morphology, growth characteristics, PCR for tissue-specific gene expression (gp100, TYRP-1, and TYRP-2) and biologic behavior, tested mycoplasma-free (MycoAlert Mycoplasma Detection Kit, Lonza), and frozen. Cells were cultured for <3 months after thawing.

B16TK cells were derived from a B16.F1 clone transfected with a plasmid expressing the Herpes Simplex Virus thymidine kinase (*HSV-1 TK*) gene[9]. Following stable selection in 1.25 μg/mL puromycin, these cells were shown to be sensitive to Ganciclovir (Cymevene) at 5 μg/ml. B16TK cells were grown in DMEM + 10% FBS (Life Technologies) + 1.25 μg/mL puromycin (Sigma).

B16-CSDE1$^{WT}$, B16-CSDE1$^{C-T}$, Hep3B-CSDE1$^{WT}$, Hep3B-CSDE1$^{C-T}$, or Mel888-CSDE1$^{WT}$ or CSDE1CSDE1$^{C-T}$ cell lines were generated by transfection of parental B16, Hep3B, or Mel888 cells with pcDNA3.1 expression vectors expressing either the murine (B16) or human (Hep3B and Mel888) CSDE1 wild-type (non-mutated) or CSDE1$^{C-T}$-mutated genes, isolated by PCR from B16 or Hep3B cells that had escaped in vitro oncolysis by VSV-mIFNβ (B16) or VSV-hIFNβ (Hep3B) in the 21-day selection protocol[10] described below. Forty-eight hours after transfection, cells were selected in G418 (5 mg/ml B16, 3 mg/ml Hep3B, and 1 mg/ml Mel888) for 2 weeks. Overexpression of the CSDE1 proteins was confirmed in these bulk G418$^r$ populations of cells by western blotting.

VSV expressing murine IFNβ (VSV-mIFNβ), human IFNβ (VSV-hIFNβ)[25], murine CSDE1$^{WT}$, murine CSDE1$^{C-T}$, or green fluorescent protein (VSV-GFP) was rescued from the pXN2 cDNA plasmid using the established reverse genetics system in BHK cells as previously described[22,25,26,58]. In brief, BHK cells are infected with MVA-T7 at an MOI of 1. Cells are incubated at 37 °C and 5% CO₂. After 1 h, cells are transfected with pVSV-XN2 genomic VSV plasmid (10 μg), pBluescript (pBS) encoding VSV-N (3 μg), pBS encoding VSV_P (5 μg), and pBS encoding VSV L proteins (1 μg) using Fugene6 according to the manufacturer's recommendations. Cells were incubated at 37 °C and 5% CO₂ for 48 h. After 48 h, supernatant was collected and clarified by passing through a 0.2 μm filter. All transgenes were inserted between viral *G* and *L* genes using the XhoI and NheI restriction sites. VSV co-expressing murine, or human, IFNβ and CSDE1$^{WT}$ or CSDE1$^{C-T}$ were also generated by cloning the *CSDE1* genes between the viral *M* and *G* genes. Virus titers were determined by plaque assay on BHK cells or on the stated cell lines in the text.

**Mice**. Female C57BL/6 mice were obtained from The Jackson Laboratory at 6–8 weeks of age Athymic nude mice *Foxn1$^{nu/nu}$* were obtained from Envigo at 4–6 weeks of age. All mice were maintained in a pathogen-free BSL2 biohazard facility. This facility is maintained between 68 and 79 °F with humidity from 30% to 70% and on a 12 h light/dark cycle. All animal studies were approved by the Institutional Animal Care and Use Committee at Mayo Clinic.

**In vivo experiments**. All in vivo studies were approved by the Institutional Animal Care and Use Committee at Mayo Clinic. Mice were challenged subcutaneously with $2 \times 10^5$ B16 melanoma cells, in 100 μL PBS (HyClone, Logan, UT, USA). Subcutaneous tumors were treated with doses of $5 \times 10^7$ pfu of VSV delivered i.t. in 50 μL of PBS. Tumors were measured using calipers three times per week and mice were killed when tumors reached 1.0 cm in diameter. For experiments using ICB, mice received 300 μg each of anti-mouse PD-1 (clone RMP1-14), per dose i.p. (BioXCell catalog number BE0146). Control mice received 300 μg of control rat IgG (Jackson ImmunoResearch catalog number 012-000-003).

**Immune cell activation**. Spleens and lymph nodes from C57Bl/6 mice were immediately excised upon killing. Single-cell suspensions were achieved in vitro via mechanical dissociation. Red blood cells were lysed by resuspension in ammonium-chloride-potassium (ACK) lysis buffer and incubating at room temperature for 2 min. Cells were resuspended at a concentration of $1 \times 10^6$ cells/mL in Iscove's modified Dulbecco's medium (Gibco) supplemented with 5% FBS, 1% penicillin–streptomycin, and 40 μmol/L 2-Mercaptoethanol. Splenocytes were co-cultured with target cells at various effector to target ratios or with stimulating peptides as described in the text. Supernatants from co-cultures were collected and assayed for TNF-α and IFN-γ by enzyme-linked immunosorbent assay (ELISA) as per the manufacturer's instructions (Mouse TNF-α or Mouse IFN-γ ELISA Kit, OptEIA, BD Biosciences, San Diego, CA).

**In vitro selection of virus-resistant populations[10]**. B16, Hep3B, or Mel888 cells were infected for 1 h with VSV at an MOI of 0.01. Cells were washed with PBS to remove any excess virus and then incubated for 7 days. Cells were washed every 2 days to remove any dead or floating cells. After 7 days, the cells were collected and re-plated. These cells were subjected to two repeated rounds of infection and re-plating as just described. After 21 days, three total rounds of infection, the remaining virus-escaped cells were collected.

**qrtPCR and sequencing**. RNA was prepared using the QIAGEN-RNeasy-MiniKit (Qiagen, Valencia, CA) as per the manufacturer's instructions. One microgram of total RNA was reverse-transcribed in a 20 μl volume using oligo-(dT) primers using the First Strand cDNA Synthesis Kit (Roche, Indianapolis, IN). A cDNA equivalent of 1 ng RNA was amplified by PCR with gene-specific primers using glyceraldehyde 3-phosphate dehydrogenase (GAPDH) as the loading control (mgapdh sense: 5′-TCATGACCACAGTCCATGCC-3′; mgapdh antisense: 5′-TCAGCTCTGGGATGACCTTG-3′). qrtPCR was carried out using a Light-Cycler480 SYBRGreenI Master kit and a LightCycler480 instrument (Roche) according to the manufacturer's instructions. The ΔΔC$_T$ method was used to calculate the fold change in the expression level of viral RNA (P, M, IGR-M, L, G, and L-G) and GAPDH as an endogenous control for all treated samples relative to an untreated calibrator sample.

The following primers were used: P1: 5′-cctctcacca-3′; P2: 3′-gctctcagtt-5′ (120 bp fragment). M1: 5′-gatctaagtg-3′; M2 3′-catacgaggc-5′ (120 bp fragment). IGR1: 5′-actatgaaaa-3′; G1: 5′-gatataagtt tcctttatac-3′; G2: 3′-ggttcttttctgtctaaata-5′

(300 bp fragment). L1: 5′-attcctgaat cccgatgagc-3′; L2: 3′-taaactgcaccacctctgga-5′ (300 bp fragment).

**Sequencing of the *CSDE1* gene.** The *CSDE1* gene was sequenced using the primer 5′-TCACGAAGTGCTGCTGAAGT-3′ and aligned with NCBI Reference Sequence: NM_144901.4.

**APOBEC3 knockdown.** Four unique 29mer shRNA retroviral constructs (Origene Technologies, Rockville, MD) were used as a combination to significantly reduce the expression of murine APOBEC3 in B16 cells compared to a single, scrambled shRNA encoding retroviral construct[10]. To achieve optimal knockdown for periods of more than 2 weeks in culture, each construct was pre-packaged as retroviral particles in the GP + E86 ecotropic packaging cell line and was used to infect B16 cells at an MOI of ~10 per retroviral construct. In addition, a single, scrambled negative control, non-effective shRNA cassette was packaged and used to infect cells, to generate B16 (scrambled shRNA) cells.

Hep3B cells were infected with a retroviral vector encoding either full-length functional APOBEC3B or a mutated, non-functional form of APOBEC3B as a negative control obtained from Reuben Harris (University of Minnesota, MN)[9–11]. Infected populations were selected for 7 days in hygromycin to generate Hep3B (APOBEC3B) or Hep3B (APOBEC3B INACTIVE) cell lines, and were used for experiments as described.

**Protein expression analysis.** Cells were lysed in NP40 lysis buffer containing Pierce Protease inhibitor tablets at a final concentration of 1× (ThermoScientific). Protein lysates were quantified by bicinchonic acid (BCA) assay according to the manufacturer's instructions (Pierce, ThermoScientific). Whole tumor cell lysates, recovered from mice in vivo, were normalized by protein concentration prior to ELISA determination of IL-12 and TNF-α (OptE1A, BD Biosciences, San Diego), to ensure equal amounts of protein were assayed from tumors of different sizes. For western blot analysis of CSDE1 (89 kDa), 20 μg protein lysate was run on a 4–15% SDS-polyacrylamide gel electrophoresis (PAGE) gel, transferred to polyvinylidene difluoride (PVDF) membrane, and blotted with anti-CSDE1, a rabbit polyclonal antibody (Bethyl Laboratories, Montgomery TX, product number #A303-160A), at a dilution of 1/500, overnight at 4 °C. Membranes were washed with 0.05% Tween-20 PBS and then probed with anti-rabbit secondary antibody (1/50,000) in 5% milk. Membranes were developed with chemiluminescent substrate (Thermo Fisher Scientific). For western blot analysis of VSV M (29 kDa), 20 μg protein lysate was run on a 4–15% SDS-PAGE gel, transferred to PVDF membrane, and blotted with anti-VSV Matrix clone 23H12, a mouse monoclonal antibody (EMD Millipore Corp, Burlington MA, product number #MABF2347), at a dilution of 1/1000, overnight at 4 °C. Membranes were washed with 0.05% Tween-20 PBS and then probed with anti-mouse secondary antibody (1/10,000) in 5% milk. Membranes were developed with chemiluminescent substrate (Thermo Fisher Scientific).

**Defective interfering particle assay[10].** Hep3BP or Hep3B-CSDE1[C-T] cells were infected with VSV-hIFNβ or with VSV-IFNβ-IGR P/M[C-U] at an MOI of 0.01 and were incubated for 72 h. Supernatant was collected and either left undiluted or diluted 1 : 10 or 1 : 100 in serum-free medium. Fresh BHK cells were seeded the day before in triplicate wells and diluted viral supernatants were allowed to adsorb for 1 h. Stock VSV-hIFNβ virus was then added at an MOI of 20 and was incubated for 1 h. Cells were then washed 3× in PBS and fresh supernatant was added. Supernatant was collected 24 h after infection and was titered by plaque assay, by limiting the dilution on BHK cells.

**Human T-cell in vitro education and re-stimulation.** Peripheral blood mononuclear cells were isolated from healthy donor apheresis cones obtained through the Mayo Clinic Blood Donor Center (approved by the Division of Transfusion Medicine Research Committee at Mayo Clinic and determined to be institutional review board exempt). Informed consent was obtained from all donors for the use of their sample for research purposes. CD3+ T cells were isolated using a magnetic sorting kit (Miltenyi Biotech) and were activated using CD3/CD28 beads (Thermo Fisher). T cells were co-cultured at a ratio of 10 : 1 with CD14+ in vitro-matured dendritic cells prepared from the same donor pre-loaded with lysates from target tumor cells at a ratio of 1 : 10 Target cell lysate : DC. On days 3 and 5, tumor cell lysates were re-added to the co-culture. After 7 days of co-culture, CD3+ T cells were re-isolated using a magnetic sorting kit (Miltenyi Biotech), co-cultured with newly matured monocyte-derived dendritic cells (MoDCs), and loaded with tumor cell lysate at a ratio of 1 : 10 Target cell lysate : DC. Three days later, supernatant was collected for IFN-γ ELISA (R&D).

In separate experiments, CD3+ T cells from donor 3 were treated as above for 7 days and were re-isolated by magnetic sorting. Target tumor cells (10^4; Hep3B parental or Hep3B-VSV-hIFNβ 21d ESC) were treated for 24 h with hIFN-γ (200 U/mL for 12 h) and then co-cultured with 10^5 of the previously primed T cells (primed/expanded on either Hep3B parental or Hep3B-VSV-hIFNβ 21d ESC cells) (triplicate wells per treatment). A further 10^5 T cells were added after 48 h. At 120 h post co-culture, wells were washed 3× with PBS and the surviving adherent cells were counted.

Autologous MoDCs were isolated using CD14+ magnetic sorting (Miltenyi Biotech). Isolated MoDCs were incubated with human granulocyte-macrophage colony-stimulating factor (GM-CSF; 800 U/mL) and IL-4 (1000 U/mL) to induce maturation. On Days 3 and 5, media was replaced with fresh human GM-CSF (1600 U/mL) and IL-4 (1000 U/mL). On Day 7, non-adherent cells were collected, washed with PBS, and resuspended in medium containing GM-CSF (800 U/mL), IL-4 (1000 U/mL), TNF-α (1100 U/mL), IL-1β (1870 U/mL), IL-6 (1000 U/mL), and PGE2 (1 μg/mL), and were incubated for 2 days. Two days later, dendritic cells were collected for co-incubation with freshly isolated, or pre-activated, T cells at a ratio of 1 : 10 as described above.

**siRNA knockdown of CSDE1.** Target cells were transfected with no siRNA, 600 pmol of Silence select Negative siRNA, or with 600 pmol of [s15373 + 15374 siRNA] (2 CSDE1-specific siRNA)[46] and levels of CSDE1 assayed by western blotting 24 or 48 h later.

**Statistical analyses.** All analysis was performed within GraphPad Prism software (Graphpad). Multiple comparisons were analyzed using one-way or two-way analysis of variances with a Tukey's post hoc multi-comparisons test. Survival data were assessed using the log-rank test using a Bonferroni correction for multiple comparisons. Data are expressed as group mean ± SD.

**Reporting summary.** Further information on research design is available in the Nature Research Reporting Summary linked to this article.

## Data availability
All data associated with this study are available within the Article, Supplementary Information, or available from the corresponding authors upon reasonable request. Source data are provided with this paper.

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

## Acknowledgements

We thank Toni L. Woltman for expert secretarial assistance. Funding provided by the National Institutes of Health (P50 CA210964 and R01CA108961 to R.V.), The European Research Council, The Richard M. Schulze Family Foundation, Mayo Foundation, Cancer Research UK, Shannon O'Hara Foundation, and Hyundai Hope on Wheels. The salaries of A.L.H. and C.B.D. were supported in part by grant T32 (AI132165) from the National Institutes of Health.

## Author contributions

Conception and design: T.K., J.T., L.E., J.P., A.M., and R.G.V. Development of methodology: T.K., J.T., L.E., V.A.J., C.B.D., A.L.H., B.Z., A.M., and R.G.V. Acquisition of data: T.K., J.T., C.B.D., A.L.H., B.Z., J.M.T., P.W., and M.R.S. Analysis and interpretation of data: T.K., J.T., L.E., C.B.D., A.L.H., A.S., P.S., M.J.B., H.P., K.H., A.M., and R.G.V. Writing, review, and/or revision of the manuscript: T.K., J.T., L.E., V.A.J., C.B.D., A.L.H., B.Z., J.P., A.S., P.S., L.I., M.M., L.R.R., M.J.B., H.P., K.H., A.M., and R.G.V. Artistic renderings: L.E. Study supervision: A.M. and R.G.V.

## Competing interests

The authors declare no competing interests.

**Additional information**

