## [Peer Review File · Nature Communications]

REVIEWER COMMENTS

Reviewer #1 (Remarks to the Author): with expertise in molecular virology and oncolytic virotherapy

This is an excellent and well-written manuscript. It shows very interesting data and novel mechanisms, including the identification of the novel positive host factor of VSV replication (CSDE1), novel mechanisms of co-evolution of cancer cells and oncolytic VSV, and very smart approaches to drive and trap tumors into an escape phenotype, which then can be targeted by vaccination against neo-antigen (such as mutant CSDE1).

I have several suggestions:

1. Authors suggest that CSDE1 specifically recognizes the intergenic region between the P and M genes (5'-aaaaa(aaGua)-3'). However, they do not mention that exactly the same sequence is also present between other genes in VSV genome (N and P, G and L). Why then CSDE1 specifically affects mRNA synthesis termination between P and M, but not between other viral genes???
2. The assay for DI particles is not sufficient to make author's claim! I suggest removing the data and claims about DI particles, or provide additional molecular data.
3. There is not enough background information in the manuscript about known functions of CSDE2, including its possible role in replication of other viruses (I found several papers).
4. The title is too general and not informative.
5. The introduction is too long, especially the last paragraph basically listing all the results.
6. A brief explanation should be added why Hep3B and B16 cell lines were chosen for most experiments.
7. I suggest removing the claim about dominant-negative role of CSDE-P5S, unless the authors would add additional experimental data demonstrating the mutant acts this way not only when it is overexpressed.
8. The authors claim in several places that 6 h post infection "represents a single cycle of replication". Without any proof, I suggest changing language to "early stage of viral replication". If authors can support this claim about "single cycle of replication", it should be added to the manuscript.
9. I suggest changing language (line 278) "VSV... generated therapy"...

Reviewer #2 (Remarks to the Author): with expertise in cancer immunotherapy

NCOMMS-20-29131A-Z

Kottke et al describe a vaccination strategy in the B16 melanoma model in which vaccination with an IFN-beta over-expressing VSV + mutated CSDE1 induced potent T cell responses against escaped tumor cells expressing a mutated CSDE1. The authors conclude that treatment with VSV-IFN-beta treatment could be improved by targeting the mutated escape tumors with immunotherapy including both heteroclitic antigen and checkpoint blockade. The therapeutic implication for this treatment strategy is compelling, as well as the evidence for the role of CSDE1 in mediating VSV replication and tumor escape.

Major comments:

1. The authors directly vaccinate with viruses expressing WT or mutant antigen, rather than treating tumors that escape after frontline VSV-IFN β , as it is suggested that this treatment could be used clinically. The discussion would benefit from a statement concerning whether other compensatory mutations could develop under the selective pressure of a vaccine in this context. If the mice vaccinated with CSDE mutant viruses are allowed to go beyond 50 days, do other escape variants develop?

2. One potential limitation of this system in that developing strong T cell responses against the escaped tumor cells is dependent on expression of the antigen and replication of VSV, which is impaired after virotherapy. Even 90% penetrance leaves some tumor cells expressing WT CSDE1. This conundrum is addressed by the development of the heteroclitic response and cross-reactivity of T cells against the neoantigen and discussed on line 265 of the results. However, given the importance of that cross-reactivity for the therapeutic implications of this approach, the manuscript would be strengthened with a direct demonstration that T cells raised against mutant CSDE1 are therapeutically sufficient to treat tumors expressing WT antigen, especially given the observation that the heteroclitic response is weaker (line 264). This could be shown with adoptive transfer of T cells raised against mutant CSDE1 into mice bearing tumors treated with VSV-IFN β , or vaccination of mice implanted with a mixture of tumor cells expressing WT or mutant antigen. This issue is made even more apparent in Fig 6, where the T cells primed against VSV-IFN β escape tumor lysates have little cytotoxicity against the parental H3B cells (the text says "some" on line 304, but the figure looks like the difference in cells remaining between no T cells and +VSV-IFN β primed T cells not significant).

3. Is checkpoint blockade required for the survival advantage of VSV-mIFN β -CSDE1(C-T)? It would be beneficial to show the survival curves for the experiments performed in Figure 5a and b, so that the therapeutic benefit of vaccination alone (without a-PD-1) could be understood as well.

4. The author's discussion statement that HLA-/patient-specific EATA will need to be identified would be strengthened by a study of the HLA types of the donors included in Figure 6. Is the HLA expressed by donor 5 predicted to bind the mutant peptide, or could that explain the lack of response?

5. In Figure 6E, it's difficult to say whether the T cell responses against DCs transfected with WT or C-T plasmids is because of a lack of reactivity or because of low expression by the DCs. Was the expression of these peptides on the transfected DCs confirmed? Wouldn't be easier, and perhaps more meaningful, to use synthetic peptides for this experiment and perform a dose curve?

Minor Comments

1. Figure 5b has a typo "resimulation"

2. Please define the abbreviation for ICB in the figure legend of Figure 5c and d, or on the figure itself.

RESPONSE TO REVIEWERS

Reviewer #1:

This is an excellent and well-written manuscript. It shows very interesting data and novel mechanisms, including the identification of the novel positive host factor of VSV replication (CSDE1), novel mechanisms of co-evolution of cancer cells and oncolytic VSV, and very smart approaches to drive and trap tumors into an escape phenotype, which then can be targeted by vaccination against neo-antigen (such as mutant CSDE1).

We thank the Reviewer for these comments.

1. Authors suggest that CSDE1 specifically recognizes the intergenic region between the P and M genes (5'-aaaaa(aaGua)-3'). However, they do not mention that exactly the same sequence is also present between other genes in VSV genome (N and P, G and L). Why then CSDE1 specifically affects mRNA synthesis termination between P and M, but not between other viral genes???

The Reviewer is absolutely correct that the virus contains very similar Inter Genic Regions (IGR) between the N&P; P&M; M&G and G&L genes. However, the IGR sequence between the P&M genes – which is where CSDE1 specifically affects mRNA termination – does actually differ from all the other IGR in a single base. Thus, the P/M IGR is (5'-aaaaa(aaGua)-3'); however, the three remaining IGR have the sequence (5'-aaaaa(aaCua)-3'). This C-G change is only present in the P/M IGR, has been confirmed by sequencing and converts the sequence (5'-aaaaa(aaCua)-3') into a perfect CSDE1 consensus binding site (5'-aaaaa(aaGua)-3'). Therefore, we hypothesize that CSDE1 specifically affects mRNA synthesis termination between P and M, and not at other very similar IGRs of the virus, because the P/M IGR is the only sequence with the perfect CSDE1 consensus binding site. We are currently carrying out RIPseq studies to investigate whether CSDE1 binds only at the perfect CSDE1 site in the P/M IGR (and not at all at the other viral IGRs), or whether there are different levels of CSDE1 binding throughout the viral IGRs. We are also mutating all the other IGR to contain the perfect CSDE1 binding site to investigate whether this generates a virus with better replication properties (where CSDE1 can bind and increase gene expression at all viral genes).

To address the Reviewer's point here we have added the following text to the **Results** on **pages 9-10**:

In contrast to the loss of RNA for the viral M protein, coupled with significantly increased levels of *P-M* RNA (**Fig.3B**), relative levels of viral *G* and *L* RNA, as well as *G-L* RNA, were not significantly changed in Hep3B, or in B16, cells infected with VSV-IFN β -IGR P/M^{C-U}, or in either cell line over-expressing CSDE1^{P5S} infected with VSV-IFN β , compared to levels in parental cells infected by VSV-IFN- β (**Fig.3D-F**). **These data indicate that CSDE1 specifically affects mRNA synthesis termination between P and M,**

and not at other IGRs of the virus. This is consistent with the IGR between the P and M genes of VSV having a unique sequence of 5'-aaaaa(aaGua)-3' – which is a perfect CSDE1 consensus binding site. In contrast, all of the remaining viral IGR (N/P, M/G, G/L) have the similar, but distinct, sequence of 5'-aaaaa(aaCua)-3'.

And to the **Discussion** on **pages 18-19**:

In this scenario, the complementary mutation of the viral IGR P/M^{C-U} would allow mutated CSDE1^{P5S} to bind to the mutated consensus site and restore wild type relative levels of P, M, and P-M readthrough RNA. Our data suggest that CSDE1 specifically affects mRNA synthesis termination between P and M, but not between other viral genes, despite the IGR sequences of VSV being very similar. However, the IGR sequence between P&M (5'-aaaaa(aaGua)-3') differs from all the other IGR by a single base. Thus, the IGR sequence between N&P, M&G and G&L is (5'-aaaaa(aaCua)-3'). This C-G change, which is only present in the P/M IGR, converts the sequence (5'-aaaaa(aaCua)-3') into a perfect CSDE1 consensus binding site (5'-aaaaa(aaGua)-3'). Therefore, we hypothesize that CSDE1 specifically affects mRNA synthesis termination between P and M, and not at other very similar IGRs of the virus, because the P/M IGR is the only site with the perfect CSDE1 consensus binding site. Experiments are underway to determine whether CSDE1 binds only at the perfect CSDE1 site in the P/M IGR (and not at all at the other viral IGRs), or whether there are different levels of CSDE1 binding throughout the viral IGRs.

2. The assay for DI particles is not sufficient to make author's claim! I suggest removing the data and claims about DI particles, or provide additional molecular data.

As requested by the Reviewer we have removed **Figure 3H** and all of the associated text.

3. There is not enough background information in the manuscript about known functions of CSDE2, including its possible role in replication of other viruses (I found several papers).

We have added the following to the **Discussion** on **pages 16-17**:

We reasoned that such mutations may be in genes/proteins which mediate escape from innate, and adaptive, immune-mediated mechanisms of tumor clearance induced by VSV infection^{26, 27, 28, 29, 30, 31}, and/or may allow infected cells to down regulate critical steps in viral replication and thereby escape oncolysis.

CSDE1 is multi-functional RNA binding protein that regulates RNA translation^{40, 41, 42, 43, 44, 45, 46, 47}. CSDE1 has not previously been reported to be involved in regulation of VSV replication, although it has been shown to stimulate cap-independent translation initiation for several other viruses (Reviewed in ref⁴⁵). Thus, knock down of CSDE1 reduced the IRES-driven translation of both human rhinovirus (HRV) and poliovirus (PV) whilst not affecting cap-dependent translation⁵⁶. In the case of HRV-2, CSDE1 binding to viral mRNA alters its structure to facilitate the further binding of the polypyrimidine tract-

binding protein (PTB) creating a structure which is necessary for translational competency⁵⁷. Thus, CSDE1 can be viewed as an RNA chaperone which facilitates the formation of tertiary protein/RNA complexes, thereby bridging viral RNAs and proteins that cannot bind directly to each other.

4. *The title is too general and not informative.*

We have changed the title to:

Tumour Escape from Oncolytic Virotherapy Drives Neo-antigenesis Which Can Be Targeted by Immunotherapy

5. *The introduction is too long, especially the last paragraph basically listing all the results.*

We have shortened the Introduction from the 890 words of the original version to 668, including a much-reduced final paragraph.

6. *A brief explanation should be added why Hep3B and B16 cell lines were chosen for most experiments.*

We have added the following text to the **Results** on **page 5**:

B16 populations, which we had previously investigated as targets for virus-mediated treatment escape through APOBEC3 mutagenesis, selected for escape from VSV-GFP (B16-VSV-GFP-ESC) were heterogeneous for both CSDE1^{WT} and CSDE1^{C-T} (**Supplementary Fig.1A&B**).

And to **page 7**:

Multiple passage of VSV-IFN β through human Hep3B-CSDE1^{WT} (as a model of human hepatocellular cancer cells against which we are testing VSV-IFN β in clinical trials) increased replication compared to passage through Hep3BP parental cells (**Fig.1F**). In contrast, after just a single passage through Hep3B-CSDE1^{C-T} cells, titers were significantly lower than with passage through Hep3BP ($p < 0.0001$) (**Fig.1F**).

7. *I suggest removing the claim about dominant-negative role of CSDE-P5S, unless the authors would add additional experimental data demonstrating the mutant acts this way not only when it is overexpressed.*

As requested by the Reviewer, we have removed reference to CSDE1^{P5S} as a dominant negative mutation throughout the text.

8. *The authors claim in several places that 6 h post infection “represents a single cycle of replication”. Without any proof, I suggest changing language to “early stage of viral*

replication". If authors can support this claim about "single cycle of replication", it should be added to the manuscript.

As requested, we have changed 'single cycle of replication' to 'early stage of viral replication' throughout the text.

9. I suggest changing language (line 278) "VSV... generated therapy"...

On **page 13**, we have now changed 'VSV-mIFN β generated therapy, but all tumors eventually escaped (**Fig.5F**).' to 'VSV-mIFN β **prolonged survival compared to PBS**, but all tumors eventually escaped (**Fig.5F**).'

Reviewer #2:

Kottke et al describe a vaccination strategy in the B16 melanoma model in which vaccination with an IFN-beta over-expressing VSV + mutated CSDE1 induced potent T cell responses against escaped tumor cells expressing a mutated CSDE1. The authors conclude that treatment with VSV-IFN-beta treatment could be improved by targeting the mutated escape tumors with immunotherapy including both heteroclitic antigen and checkpoint blockade. The therapeutic implication for this treatment strategy is compelling, as well as the evidence for the role of CSDE1 in mediating VSV replication and tumor escape

We thank the reviewer for these comments.

Major comments:

1. *The authors directly vaccinate with viruses expressing WT or mutant antigen, rather than treating tumors that escape after frontline VSV-IFN β , as it is suggested that this treatment could be used clinically. The discussion would benefit from a statement concerning whether other compensatory mutations could develop under the selective pressure of a vaccine in this context. If the mice vaccinated with CSDE mutant viruses are allowed to go beyond 50 days, do other escape variants develop?*

The Reviewer's point is well taken. We initially targeted the CSDE1^{P5S} mutation in cells which escaped VSV-IFN β therapy because our RNAseq data showed it to be reproducibly the highest frequency mutation in ESC cells and, therefore, as close to a trunk mutation in cells escaping VSV-IFN β as we can find. We did, however, also identify other mutations in VSV-IFN β ESC tumors, but at much lower frequencies in the populations. It is possible that applying intense pressure against the CSDE1^{P5S} mutation may allow these other escape-induced mutations to become more prominent both functionally and, therefore, quantitatively in ESC populations. We did not see further escape in the mice vaccinated with the CSDE1^{P5S} mutant viruses in the experiments of **Fig.5F** but we agree with the Reviewer that that does not mean that other compensatory mutations may not develop in clinical circumstances. To address

the Reviewer's point here, we have added the following to the **Discussion** on **pages 21-22**:

Simultaneously, VSV-IFN β -CSDE1^{C-T} also provides high concentrations of target antigen (CSDE1^{P5S}) (reflected by IL-12 only in VSV-IFN- β -CSDE1^{C-T}-injected tumors, **Fig.5C**), which are lacking with i.p. DC and intra-tumoral VSV-IFN β .

We targeted the CSDE1^{P5S} mutation in cells which escaped VSV-IFN β therapy because it was the highest frequency mutation in VSV-IFN β ESC cells and, therefore, represents a 'trunk-like' mutation in cells escaping VSV-IFN β (**Supplementary Fig.1**). However, applying intense immunotherapeutic pressure against the CSDE1^{P5S} mutation, as in **Fig.5**, may allow other, lower frequency (branch-like) escape induced mutations to become more prominent in ESC populations to compensate for a requirement of ESC tumors to lose detectable expression of CSDE1^{P5S} completely. These mutations may be in cellular proteins/pathways affecting, for example, viral replication, the anti-viral response and/or antigen presentation.

2. One potential limitation of this system in that developing strong T cell responses against the escaped tumor cells is dependent on expression of the antigen and replication of VSV, which is impaired after virotherapy. Even 90% penetrance leaves some tumor cells expressing WT CSDE1. This conundrum is addressed by the development of the heteroclitic response and cross-reactivity of T cells against the neoantigen and discussed on line 265 of the results. However, given the importance of that cross-reactivity for the therapeutic implications of this approach, the manuscript would be strengthened with a direct demonstration that T cells raised against mutant CSDE1 are therapeutically sufficient to treat tumors expressing WT antigen, especially given the observation that the heteroclitic response is weaker (line 264). This could be shown with adoptive transfer of T cells raised against mutant CSDE1 into mice bearing tumors treated with VSV-IFN β , or vaccination of mice implanted with a mixture of tumor cells expressing WT or mutant antigen. This issue is made even more apparent in Fig 6, where the T cells primed against VSV-IFN β escape tumor lysates have little cytotoxicity against the parental H3B cells (the text says "some" on line 304, but the figure looks like the difference in cells remaining between no T cells and +VSV-IFN β primed T cells not significant).

In response to the Reviewer's point, we have added a new **Supplemental Figure 3**. These new data show directly that T cells raised against mutant CSDE1^{P5S} (from mice treated with VSV-CSDE1^{P5S} which rejected ESC tumors, and expanded *in vitro* with CSDE1^{P5S} peptide) are therapeutically sufficient to treat B16 tumors (expressing only WT CSDE1^{WT} antigen). Therefore, we have added the following text in the **Results** on **pages 13-14**:

However, expression of the CSDE1^{P5S} EATA from the virus completely prevented tumor escape (**Fig.5F**), despite significantly less replication in tumors compared to either VSV-mIFN β or VSV-mIFN β -CSDE1^{WT} (**Figs.5G&4**). It is highly unlikely that evolution of the escape promoting CSDE1^{P5S} mutation occurs in 100% of all cells in the ESC tumors (e.g. **Supplementary Fig.1J**). Therefore, our model of tumor clearance depends upon the heteroclitic anti-CSDE1^{P5S}/anti-CSDE1^{WT} T cell response being potent enough to

clear that proportion of tumor cells in which the CSDE^{P5S} mutation had not evolved following VSV-IFN β therapy. In this respect, as we have seen previously, adoptive transfer of *in vitro* activated OT-I CD8+ T cells (specific for the irrelevant SIINFEKL epitope of Ovalbumin) in combination with anti-PD-1 ICB, had no significant therapeutic effect upon the growth of subcutaneous B16 tumors (100% CSDE^{WT}) (**Supplementary Fig.3A**). In contrast, CD8+ T cells recovered from mice which had survived B16 tumors treated with VSV-CSDE^{P5S} (**Fig.5F**), and expanded *in vitro* against the mutated CSDE^{P5S} MFSDSNLLH peptide, significantly extended survival compared to the control treated group, and cured a proportion of mice (**Supplementary Fig.3A**). Addition of ICB with anti-PD-1 antibody significantly further enhanced the efficacy of the adoptive transfer of anti-CSDE^{P5S} CD8+ T cells, and cured 100% of mice (**Supplementary Fig.3A**). Finally, adoptive transfer of anti-CSDE^{P5S} CD8+ T cells in combination with frontline treatment with VSV-IFN β also cured all the mice (even in the absence of ICB), whereas a combination of VSV-IFN β and OT-1 CD8+ T cells was no more effective than virus alone (**Supplementary Fig.3B**). These data show that T cells raised against mutant CSDE^{P5S} are therapeutically sufficient to treat tumors expressing CSDE^{WT} antigen, despite the weaker strength of the heteroclitic response against B16 cells compared to that against B16-CSDE^{P5S} expressing ESC cells.

And we have added new text to the **Discussion** on **page 21**:

VSV-IFN β -ESC tumors *in vivo* rarely contained a completely homogenous population of CSDE^{C-T} mutant tumor cells (**Supplementary Fig.1J**). Therefore, the heteroclitic anti-CSDE^{P5S} T cell responses¹¹ (**Fig.5B**) probably contribute a significant bystander effect against tumor cells which do not become infected, escape direct oncolysis or innate immune clearance, or which do not evolve the CSDE^{C-T} mutation. This model is supported by the data in **Supplementary Fig.3**, in which adoptive transfer of anti-CSDE^{P5S} CD8+ T cells improved survival of mice bearing B16 (CSDE^{WT}) tumors (**Supplementary Fig.3A**) or cured them when used in combination with either anti-PD-1 ICB (**Supplementary Fig.3A**) or with frontline, CSDE^{P5S}-inducing VSV-IFN β therapy (without ICB) (**Supplementary Fig.3B**).

And a new **Supplemental Figure 3 Legend**:

Supplemental Figure 3: C57Bl/6 mice with 10d established s.c. B16 tumors (7 mice/grp) were treated **A.** i.v. with 2.5×10^6 OT-1 T cells activated *in vitro* for 5d with IL-2 (50 IU rhIL-2/ ml) and SIINFEKL peptide (1 μ g/ml) (d10), and with anti-PD-1 antibody i.p. (300 μ g/injection) (d17,19,21); or with 2.5×10^6 CD8+ T cells recovered from the spleens of mice which had rejected B16 tumors treated with VSV-CSDE^{P5S} (**Fig.5F**) and activated *in vitro* for 5d with IL-2 and the CSDE^{P5S} mutated peptide MFSDSNLLH (1 μ g/ml) (d10) either with, or without, anti-PD-1 antibody (d17,19,21). **B.** 3 additional groups were treated intratumorally with VSV-IFN- β (d10,12,14) (5×10^7 pfu/injection) and subsequently with either CD8+ T cells from naive C57Bl/6 mice (d17); *in vitro* activated OT-I CD8+ T cells; or *in vitro* activated anti-CSDE^{P5S} CD8+ T cells. Survival with time is shown.

3. Is checkpoint blockade required for the survival advantage of VSV-mIFN β -CSDE1(C-T)? It would be beneficial to show the survival curves for the experiments performed in Figure 5a and b, so that the therapeutic benefit of vaccination alone (without a-PD-1) could be understood as well.

We do not have survival curves for **Fig.5A&B** because the experiment was intentionally stopped at Day 30 as the first animals had to be euthanized due to tumor size (in order to harvest splenocytes for the re-stimulation assays shown in **Fig.5B**). However, at that point (day 30), there was a significant difference in tumor volumes between mice treated with VSV-CSDE1^{P5S} compared to VSV-CSDE1^{WT}. We have added these data as part of a new **Fig. 5A**. These data show that, even in the absence of anti-PD-1 ICB therapy, VSV-IFN β -CSDE1^{P5S} confers better anti-tumor activity than VSV-IFN β -CSDE1^{WT}, despite the fact that VSV-CSDE1^{WT} replicates more efficiently.

In addition, we have added a new **Supplemental Figure 3**, as described in response to *Point 2* above. These data show that anti-CSDE1^{P5S} CD8 T cells have anti-tumor activity against B16 tumors even in the absence of anti-PD-1 ICB (eg **Supplementary Fig.3A**) and that they are active against B16 tumors which escape VSV-IFN β therapy without anti-PD-1 ICB, at least in the context of *in vivo* activation and adoptive T cell transfer (**Supplementary Fig.3B**).

Therefore, to address the Reviewer's point here, we have added the following text to the **Results** on **page 12**:

Although VSV-mIFN β -CSDE1^{WT} did not generate α -CSDE1^{WT} T cells, VSV-mIFN β -CSDE1^{C-T} induced potent T cell responses against the CSDE1^{P5S} neoantigen (**Fig.5B**), as well as weaker responses against B16-CSDE1^{WT}, and B16 (expressing endogenous CSDE1), confirming that CSDE1^{P5S} acts as a heteroclitic neo-epitope in the C57Bl/6 model¹¹. **These T cell responses probably contributed to the significantly reduced tumor sizes in mice treated with VSV-mIFN β -CSDE1^{C-T} compared to those treated with VSV-mIFN β -CSDE1^{WT} at day 30 when this experiment was stopped (Fig.5A).**

And to the **Discussion** on **page 20**:

VSV-IFN β -CSDE1^{C-T}, replicated significantly less well than VSV-IFN β or VSV-IFN β -CSDE1^{WT} (**Figs.4B&5G**), but induced potent T cell responses against the CSDE1^{P5S} EATA (**Fig.5B**), which completely prevented escape (**Fig.5F**) **in the presence of anti-PD-1 ICB. Anti-CSDE1^{P5S} T cells still had anti-tumor efficacy without ICB as evidenced by the significant reduction in tumor volumes in Fig.5A. In addition, anti-CSDE1^{P5S} CD8 T cells had anti-tumor activity against B16 tumors even in the absence of anti-PD-1 ICB (eg Supplementary Fig.3A) and were active against B16 tumors which escaped VSV-IFN β therapy without anti-PD-1 ICB, at least in the context of *in vivo* activation and adoptive T cell transfer (Supplementary Fig.3B).** Although VSV-mIFN β -CSDE1^{WT} was a significantly better oncolytic than VSV-IFN β (**Figs.4B&C&5F&G**), it did not generate α -CSDE1^{WT}, or α -CSDE1^{P5S}, T cell responses (**Fig.5B**), suppressed evolution of the CSDE1^{P5S} immunogen in escaping cells (**Fig.4D**) and was not as effective as VSV-IFN β -CSDE1^{C-T} (**Fig.5F**). Thus, the therapeutic value of T cell control of emerging

escape variants outweighed the loss of oncolytic potency of VSV-IFN β -CSDE1^{C-T} (**Fig.5F**).

And a new Legend to **Figure 5A**:

A. C57Bl/6 mice bearing 10d B16 tumors were injected i.t. with PBS, VSV-mIFN β , VSV-mIFN β -CSDE1^{WT} or VSV-mIFN β -CSDE1^{C-T}. At day 30, mice were euthanized for harvesting of splenocytes and tumor sizes measured as shown.

4. The author's discussion statement that HLA-/patient-specific EATA will need to be identified would be strengthened by a study of the HLA types of the donors included in Figure 6. Is the HLA expressed by donor 5 predicted to bind the mutant peptide, or could that explain the lack of response?

5. In Figure 6E, it's difficult to say whether the T cell responses against DCs transfected with WT or C-T plasmids is because of a lack of reactivity or because of low expression by the DCs. Was the expression of these peptides on the transfected DCs confirmed? Wouldn't be easier, and perhaps more meaningful, to use synthetic peptides for this experiment and perform a dose curve?

We apologize but we did not have access to the HLA expression of Donor 5 so we were unable to assess if the binding predictions in **Figure 6D** correlated with the lack of response.

We confirmed efficient transfection of the DC by surrogate expression of GFP positivity (greater than 60% GFP CD14+ positivity was used as the quality control for these experiments) but we did not confirm peptide expression as we do not have an appropriate antibody. However, as requested by the Reviewer, we have now added an additional **Figure 6F** in which we show that CD3+ T cells across three additional donors could be stimulated to secrete IFN- γ in response to peptides which contain the CSDE1^{P5S} mutation, but not in response to peptides containing the wild type CSDE1 sequence.

Therefore, we have added the following test to the **Results** on **pages 15-16**:

Both donors 4&5 showed high level T cell priming against Hep3B-VSV-IFN β -ESC cells compared to Hep3B (**Fig.6E**), as with **Fig.6B**. Thus, escape from VSV-hIFN β generated cells which were consistently more immunogenic than parental in both human and murine contexts.

CD3+ T cells from one of three additional donors secreted significantly more IFN- γ when stimulated *in vitro* with DC-presented, mutated CSDE1^{P5S} 9mer peptide (MSFDSNLLH) compared to either the DC-presented wild type CSDE1 9mer (MSFDPNLLH) or the negative control SIINFEKL peptide (**Fig.6F**). However, CD3+ T cells from all three donors secreted significantly more IFN- γ when stimulated *in vitro* with a 20mer peptide in which the Pro-Ser mutation in CSDE1^{P5S} could potentially be presented by DC at any position in a loaded HLA molecule, compared to when DC were loaded with the wild type 20mer or the SIINFEKL peptide (**Fig.6F**). These data provide additional support for the hypothesis that neo-antigenesis of hCSDE1^{P5S} serves as an EATA.

And a new **Figure Legend** to **Figure 6**:

F. Human CD3+ T cells activated *in vitro* and co-cultured with autologous DC were cultured with DC pulsed 48hrs previously with 5µg/ml peptide. Pulsed DC, were re-added on d5. On d7 isolated CD3+ T cells were co-cultured with similarly pulsed DC. 72hrs later, supernatants were assayed for IFN-γ. Peptide pulsing of DC was with: Lane 1: SIINFEKL peptide from Ovalbumin; Lane 2: wild type CSDE1 9mer peptide MSFDPNLLH; Lane 3: an extended CSDE1 20mer peptide, MSFDPNLLHNNGHNGYPNGT, which could, in theory, be processed into any 9-11mer in which the Pro at position 5 could be at different positions; Lane 4: mutated CSDE1^{P5S} MSFDSNLLH peptide; or Lane 5: an extended CSDE1^{P5S} 20mer peptide, MSFDSNLLHNNGHNGYPNGT, in which the Ser at position 5 could be at different positions.

Minor Comments

1. *Figure 5b has a typo “resimulation”*

Thank you; We have corrected the typo.

2. *Please define the abbreviation for ICB in the figure legend of Figure 5c and d, or on the figure itself.*

We have added the definition to the Figure Legend of Fig.5C.

REVIEWERS' COMMENTS

Reviewer #1 (Remarks to the Author):

The revised and much-improved manuscript has successfully addressed all my comments!

Reviewer #2 (Remarks to the Author):

All comments have been adequately addressed.